# Revisit Multimodal Meta-Learning through the Lens of Multi-Task Learning

**Milad Abdollahzadeh, Touba Malekzadeh, Ngai-Man Cheung**
Singapore University of Technology and Design
{milad_abdollahzadeh, touba_malekzadeh, ngaiman_cheung}@sutd.edu.sg

## Abstract

Multimodal meta-learning is a recent problem that extends conventional few-shot meta-learning by generalizing its setup to diverse multimodal task distributions. This setup makes a step towards mimicking how humans make use of a diverse set of prior skills to learn new skills. Previous work has achieved encouraging performance. In particular, in spite of the diversity of the multimodal tasks, previous work claims that a single meta-learner trained on a multimodal distribution can sometimes outperform multiple specialized meta-learners trained on individual unimodal distributions. The improvement is attributed to knowledge transfer between different modes of task distributions. However, there is no deep investigation to verify and understand the knowledge transfer between multimodal tasks. Our work makes two contributions to multimodal meta-learning. First, we propose a method to *quantify knowledge transfer* between tasks of different modes at a micro-level. Our quantitative, task-level analysis is inspired by the recent transference idea from multi-task learning. Second, inspired by hard parameter sharing in multi-task learning and a new interpretation of related work, we propose a *new multimodal meta-learner* that outperforms existing work by considerable margins. While the major focus is on multimodal meta-learning, our work also attempts to shed light on task interaction in conventional meta-learning. The code for this project is available at https://miladabd.github.io/KML.

## 1 Introduction

**Multimodal meta-learning** was recently proposed as an extension of conventional few-shot meta-learning. In [1], it is defined as a meta-learning problem that involves classification tasks from multiple different input and label domains. An example in their work is a 3-mode few-shot image classification which includes tasks to classify characters (Omniglot) and natural objects of different characteristics (FC100, mini-ImageNet). The multimodal extension of meta-learning is proposed with two objectives. First, it generalizes the conventional meta-learning setup for more diverse task distributions. Second, it makes a step towards mimicking humans' ability to acquire a new skill via prior knowledge of a set of diverse skills. For example, humans can quickly learn a novel snowboarding trick by exploiting not only fundamental snowboarding knowledge but also skiing and skateboarding experience [1].

Multimodal Model-Agnostic Meta-Learning (MMAML) [1] proposes a framework to better handle multimodal task distributions and achieves encouraging performance. As one of the most intriguing findings, MMAML claims that a single meta-learner trained on a multimodal distribution can sometimes outperform multiple specialized meta-learners trained on individual unimodal distributions. This was observed in spite of the diversity of the multimodal tasks. In [1], this observation is attributed to knowledge transfer across different modes of multimodal task distribution.

35th Conference on Neural Information Processing Systems (NeurIPS 2021).

**In our work,** we delve into understanding knowledge transfer in multimodal meta-learning. While improved performance using a multimodal task distribution is reported in [1], there is no deep investigation to verify and understand how tasks from different modes benefit from each other. Towards understanding knowledge transfer in multimodal meta-learning at a micro-level, we propose a new quantification method inspired by the idea of *transference* recently proposed in multi-task learning (MTL) [2]. Despite the large number of meta-learning algorithms proposed in the literature, we remark that little work has been done in analyzing meta-learning at the task sample level. In particular, to the best of our knowledge, there is no previous work on understanding knowledge transfer among tasks and quantitative analysis of task samples in the context of meta-learning. Because of the lack of such study, the interaction between task samples remains rather opaque in meta-learning. Interestingly, despite the notion of a task in meta-learning and MTL, there has not been much intersections between these two branches of research, perhaps due to several fundamental differences between meta-learning and MTL[1] (e.g, meta-learning optimizes the risk over a large number of future tasks sampled from an unknown distribution of tasks, while MTL optimizes the average risk over a finite number of known tasks; this will be further discussed). We note that because of these fundamental differences, we propose adaptations to develop our method to quantify knowledge transfer for multimodal tasks in meta-learning.

Another contribution of our work is a new multimodal meta-learner. Our idea is inspired by *hard parameter sharing* in MTL [4]. Furthermore, we discuss our own interpretation of the modulation mechanism in [1]. These lead us to propose a new method that achieves substantial improvement over the best results in multimodal meta-learning [1]. While our major focus in this work is on multimodal meta-learning, we have also performed experiments on conventional meta-learning for our proposed knowledge transfer quantification. Our work makes an attempt to shed light on task interaction in conventional meta-learning.

Our main contributions are

- Focusing on multimodal meta-learning, we propose a method to understand and quantify knowledge transfer across different modes at a micro-level.

- We propose a new multimodal meta-learner that outperforms existing state-of-the-art methods by substantial margins.

## 2 Related Work

**Few-Shot Learning.** In a few-data regime, conventional learning methods mostly fail due to overfitting. Fine-tuning a pre-trained network [5, 6, 7, 8] sometimes prevents overfitting but at the cost of computation [9]. Therefore, recent successful approaches tackle this problem by meta-learning [10]. These methods can be classified into several categories. In *metric-based* approaches, a similarity metric between support and query samples is learned by learning an embedding space, in which samples from similar classes are close and samples from different classes are further apart [11, 12, 13, 14, 15, 16]. *Optimization-based* methods focus on learning an optimizer, including an LSTM meta-learner for replacing stochastic gradient descent optimizer [17], a mechanism to update weights using an external memory [18], or finding a good initialization point for model parameters for fast adaptation [19, 20, 21, 22]. *Augmentation-based* methods learn a generator from the existing labeled data to further use it for data augmentation in novel classes [23, 24]. Finally, *weight-generation* methods directly generate the classification weights for unseen classes [25, 26].

**Multi-Task Learning.** MTL algorithms can generally be divided into *hard* or *soft parameter sharing* methods [4]. Hard parameter sharing is the most prevalent design to MTL where a subset of the hidden layers are shared among all tasks, while several task-specific layers are stacked on top of the shared base [27]. Hard parameter sharing reduces the risk of overfitting and enables parameter efficiency across tasks [3, 28]. In soft parameter sharing, each task has a separate model. Then, either the parameters of the models are encouraged to be similar by regularizing a distance [29, 30] or the knowledge of tasks are linearly combined to produce output for each task [31]. Other works try to improve the multi-task performance by addressing *what to share* [32, 33, 34], *which tasks to train together* [35, 36], or *inferring task-specifc model weights* [37, 30, 38]. Recently, the *transference* is

---

[1]Here we follow the MTL definition in Zhang and Yang [3] instead of a loose definition of MTL: learning problems that involve more that one task.

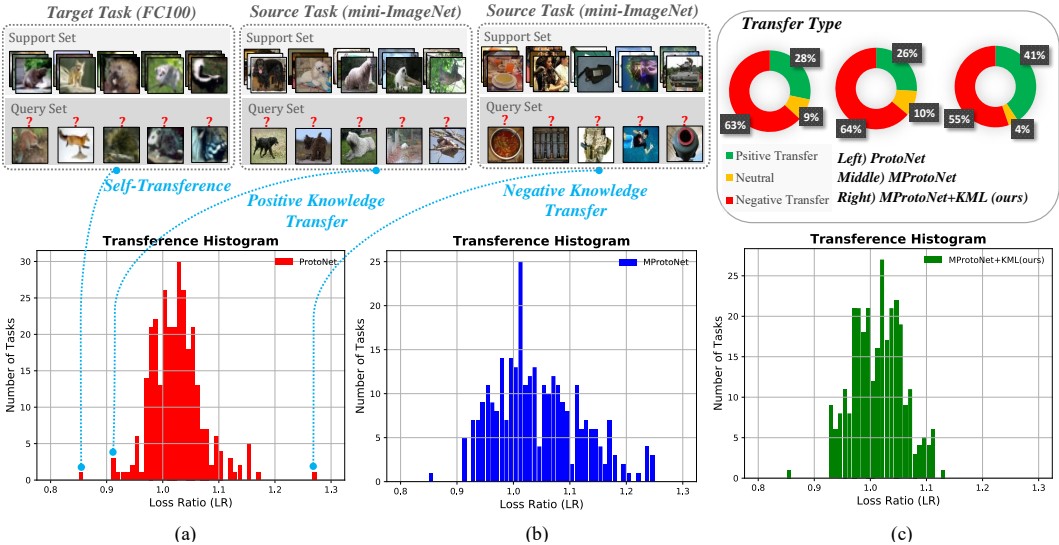

Figure 1: Information transfer (transference) from 300 meta-train mini-ImageNet tasks to a meta-test FC100 task. Transference Histogram for: (a) ProtoNet, (b) MProtoNet (with FiLM modulation), (c) MProtoNet with proposed KML method. For both positive knowledge transfer ($LR < 1$), and negative knowledge transfer ($LR > 1$) an exemplar task is shown. Proposed method increases the positive transfer from average of 27% to 41%. Here, we simply use the $LR$ threshold to classify the transference of a task as positive or negative.

proposed to analyze the information transfer in a general MTL framework [2]. This metric is then used for selecting the tasks for updating shared parameters.

## 3 Preliminaries

In the *N-way*, *K-shot* image classification task, given *N* classes and *K* labeled examples per class, the final goal is to learn a model that can generalize well to unseen examples from the same classes. Each task $\mathcal{T}$ consists of a *support* set $\mathcal{S}_{\mathcal{T}} = \{(x_i, y_i)\}_{i=1}^{N \times K}$ and a query set $\mathcal{Q}_{\mathcal{T}} = \{(\tilde{x}_i, \tilde{y}_i)\}_{i=1}^{N \times M}$. The model learns from the samples in the support set $\mathcal{S}_{\mathcal{T}}$ and is evaluated on the query set $\mathcal{Q}_{\mathcal{T}}$. As a solution, meta-learning assumes that tasks are sampled from a distribution of tasks $p(\mathcal{T})$, and are split into meta-training and meta-test sets. The meta-learner learns the prior knowledge about the underlying structure of tasks using the meta-training set, such that later, the meta-test tasks can benefit from this prior knowledge. While early meta-learning algorithms directly minimize the average training error of a set of training tasks, Vinyals et al. [12] propose a novel training strategy called *episodic training*. Episodic training utilizes sampled mini-batches called *episodes* during training, where each episode is designed to mimic the few-shot task and generated by subsampling of classes and samples from the training dataset. The use of episodes improves the generalization by making the training process more faithful to the test environment and has become a dominant approach for the few-shot learning [39, 19, 11]. Meta-learning algorithms usually suppose the task distribution $p(\mathcal{T})$ is *unimodal*, meaning that all generated classification tasks belong to a single input-label domain (e.g. classification tasks with different combination of digits). Then, a *multimodal* counterpart can be considered where it contains classification tasks from multiple different input-label domains (e.g. few-shot digits classification and birds classification). In our work, multimodal meta-learning refers to the multimodality occurs in task distribution $p(\mathcal{T})$ due to using tasks from multiple domains. This should not be confused with multimodality in data type [40] (e.g., combination of image and text).

## 4 Information Transfer Among Tasks

**Transference in Episodic Training.** The episodic training process can be viewed from a multi-task learning point of view, where multiple tasks (episodes) collaborate to build a shared feature

**Algorithm 1:** Measuring Transference on a Target Task.

**Require:** task distribution $p(\mathcal{T})$, learning rate $\alpha$, current state of network parameters $\theta^t$

1 Sample a batch of tasks $\xi = \{\mathcal{T}_i\} \sim p(\mathcal{T})$
2 Sample a target task $\mathcal{T}_j$
3 Use support set of target task $\mathcal{S}_j$ for adaption
4 Use query set $\mathcal{Q}_j$ to evaluate the adapted model and compute the loss $\mathcal{L}_{\mathcal{T}_j}(\mathcal{Q}_j; \theta^t, \mathcal{S}_j)$
5 **for** *all* $\mathcal{T}_i \in \xi$ **do**
6      Use support set $\mathcal{S}_i$ to adapt to that task
7      Use query set $\mathcal{Q}_i$ to evaluate the adapted model and compute the loss $\mathcal{L}_{\mathcal{T}_i}(\mathcal{Q}_i; \theta^t, \mathcal{S}_i)$
8      Update model parameters with respect to task $i$ using (1)
9      Compute the loss of target task using updated parameters $\mathcal{L}_{\mathcal{T}_j}(\mathcal{Q}_j; \theta_i^{t+1}, \mathcal{S}_j)$
10      Compute transference from task $i$ to target task $LR_{i \rightarrow j}$ using (2)
11 **end**

---

representation that is broadly suitable for many tasks. Consequently, training episodes implicitly transfer information to each other by updating this shared feature representation with successive gradient updates. Then, the information transfer (transference) in episodic training can be viewed as the effect of gradients update from one episode (or a group of episodes) to the network parameters on the generalization performance on other episodes. Like MTL, in an episodic training scenario, some learning episodes can be constructive regarding a target task and some can be destructive [41].

**Analysis of Transference.** To gain insights on the interaction between different episodes during episodic training, we adapt the *transference* idea [2] from multi-task learning to episodic training scenario of meta-learning. Consider a meta-learner parameterized by $\theta$. At time-step $t$ of episodic training, a batch of tasks are sampled from task distribution $p(\mathcal{T})$. Then for task $i$ denoted by $\mathcal{T}_i$, the model uses its support set $\mathcal{S}_i$ to adapt to it, and the quality of adaption is evaluated by the loss on query set $\mathcal{Q}_i$ denoted by $\mathcal{L}_{\mathcal{T}_i}$. The quantity $\theta_i^{t+1}$ is defined as updated model parameters after a SGD step with respect to task $i$:

$$\theta_i^{t+1} = \theta^t - \alpha \nabla_{\theta^t} \mathcal{L}_{\mathcal{T}_i}(\mathcal{Q}_i; \theta^t, \mathcal{S}_i) \tag{1}$$

We can use this quantity to calculate a loss which reflects the effect of task $i$ on the performance of others. More specifically, in order to assess the effect of the gradient update of task $i$ on a given target task $j$, we calculate the ratio between loss of task $j$ before and after applying the gradient update on the shared parameters with respect to $i$:

$$LR_{i \rightarrow j} = \frac{\mathcal{L}_{\mathcal{T}_j}(\mathcal{Q}_j; \theta_i^{t+1}, \mathcal{S}_j)}{\mathcal{L}_{\mathcal{T}_j}(\mathcal{Q}_j; \theta^t, \mathcal{S}_j)} \tag{2}$$

In episodic training, the loss ratio $LR_{i \rightarrow j}$ can be considered as a measure of transference from meta-train task $i$ to meta-test task $j$. If $LR_{i \rightarrow j}$ has a value smaller than one, the update on the network parameters results in a lower loss on target task $j$ than the original parameter values, meaning that task $i$ has a constructive effect on the generalization of the model on task $j$. On the other hand, if $LR_{i \rightarrow j}$ has a value greater than one, it indicates the destructive effect of task $i$ on the target task. For $i = j$, the loss ratio denotes the *self-transference*, i.e. the effect of a task's gradient update in its own loss which can be used as a baseline for transference. One major limitation of the transference algorithm proposed in [2] is that while the knowledge transfer is measured by generalization performance, they have only considered the LR improvement on the target training tasks. This can not necessarily guarantee the improvement in the generalization performance. We adrress this by sampling source tasks from meta-train dataset and target tasks from meta-test dataset. Then, our transference metric can be considered as a generalization metric in micro-level. So we expect the algorithms with better generalization to have a more positive knowledge transfer in terms of our transference metric, and vice versa. The overall procedure of calculating transference within an episodic training framework is summarized in Algorithm 1.

**Experiments.** To investigate the interaction between tasks during training in a multimodal task distribution, we have sampled 300 meta-train tasks from the mini-ImageNet dataset as source tasks.

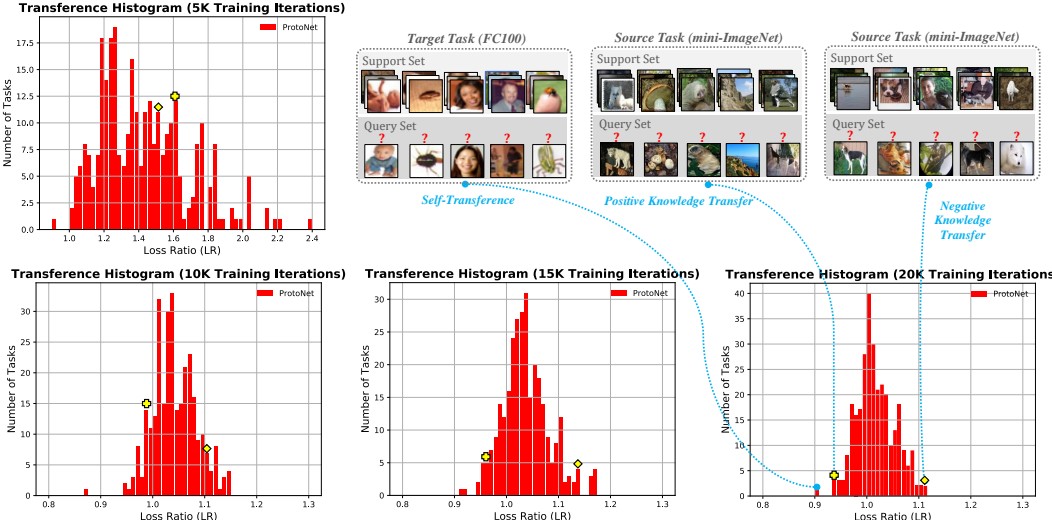

Figure 2: Transference from 300 mini-ImageNet meta-train tasks to a FC100 meta-test task. In the beginning of the training, when network learns low-level features, both tasks transfer negative knowledge. As the training proceeds, one becomes more and more positive, and then, consistently transfers positive knowledge to target task.

Then we have analyzed the transference from these tasks to a single FC100 meta-test task using algorithm 1. For this experiment, we have used ProtoNet [11] as meta-learner, and the analysis is performed in the middle of training on the combination of mini-ImageNet and FC100 datasets. The details of the experimental setup can be found in the supplementary. The histogram of the transference is shown in Figure 1a which indicates both positive and negative knowledge transfer from mini-ImageNet tasks to the target task. An exemplar task for both positive and negative knowledge transfer is shown in figure 1. The task including animal classes has positive knowledge transfer to target task while the task including non-animal classes has negative transfer.

In figure 2, the transference to a different meta-test FC100 target task from mini-ImageNet meta-train tasks is shown. While the target task includes classification from people and insect classes, two source tasks with animal classes are among the best and worst knowledge transferring source tasks. This can be attributed to the quality of samples in these tasks. When a task includes noisy data samples, it is much harder to solve meaning that the transference can also happen based on task hardness [42]. Figure 2 also indicates that in the cross mode knowledge transfer, the negative transference occurs at the beginning iterations and increasingly more positive transference occurs as training proceeds. Based on the experience from MTL literature, the negative knowledge transfer occurs when different tasks fight for the capacity [41]. In the next section we will propose a new modulation scheme to reduce negative transfer and improve generalization (Figure 1c).

## 5 Proposed Multimodal Meta-Learner

In the previous section, the transference analysis shows that different tasks in episodic training can have a positive or negative impact on the learning process of other tasks. Recent works have analyzed the compatibility of tasks in multi-task learning [36, 2] during training to select some grouping of tasks for co-training that can improve performance. In our episodic training scenario, ideally, we would like to select the grouping of learning episodes that are compatible with the meta-test task. However, direct application of the methods in MTL to episodic training is not possible, due to two main reasons. *First*, the novel tasks in meta-testing are unseen and unknown during meta-training. *Second,* even if the tasks in meta-testing were known, it can be very computationally expensive to determine the group of cooperative tasks, and then assign a task-specific layer for each task, because episodic training involves tens of thousands of training tasks. Here, we propose a new interpretation of the modulation mechanism in MMAML. These lead us to propose pseudo-task-specific (task-aware) layers inspired by hard parameter sharing in MTL.

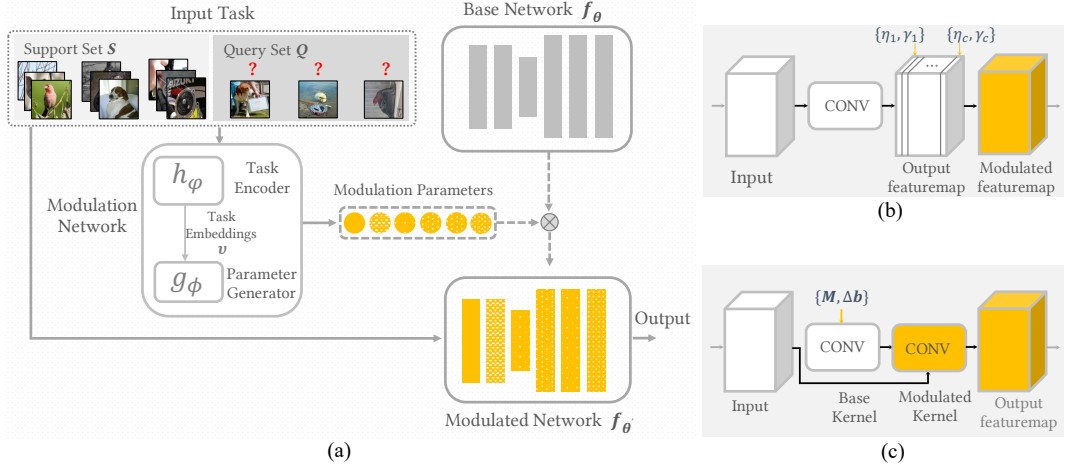

Figure 3: (a) General multimodal meta-Learning framework; Modulation scheme in (b) MMAML, (c) Proposed KML method.

## 5.1 Multimodal Model-Agnostic Meta-Learning

To tackle multimodal few-shot tasks, MMAML [1] proposes a multimodal meta-learning framework which consists of a modulation network and a base network $f_\theta$ (Figure 3a). The modulation network, predicts the mode of a task and generates a set of task-specific modulation parameters to help the base network to better fit to the identified mode. It includes a task encoder network and a modulation parameter generator. Task encoder takes the support samples of a task $\mathcal{T}$ as input and produces an embedding vector $\boldsymbol{v}_\mathcal{T} = h_\varphi(\mathcal{S}_\mathcal{T})$ to encode its characteristics. Then, task-specific modulation parameters are generated using the embedding vector of task by the parameter generator network $\boldsymbol{\omega}_\mathcal{T} = g_\phi(\boldsymbol{v}_\mathcal{T})$. Since MMAML uses feature-wise linear modulation (FiLM) [43], the generated modulation parameters are split into scaling and shifting parameters $\boldsymbol{\omega}_\mathcal{T} = \{\boldsymbol{\eta}_\mathcal{T}, \boldsymbol{\gamma}_\mathcal{T}\}$. Let $\mathbf{Y}_i$ denote the $i^{th}$ output channel of a layer in the base network $f_\theta$. Then, the corresponding modulated output channel in MMAML is calculated as:

$$\hat{\mathbf{Y}}_i = \eta_i \mathbf{Y}_i + \gamma_i \tag{3}$$

where $\eta_i \in \boldsymbol{\eta}_\mathcal{T}$ and $\gamma_i \in \boldsymbol{\gamma}_\mathcal{T}$ are scalar values. For task encoder, MMAML uses a 4-layer convolutional network which is followed by multiple MLPs as parameter generator. MMAML uses the model-agnostic meta-learning (MAML) algorithm [19] as meta-learner in base network.

**Limitations of Feature-Wise Modulation.** We believe the major limitation of MMAML results from using FiLM modulation scheme. For each convolutional layer of a CNN, the $i^{th}$ output channel (feature map) is computed by convolving the $i^{th}$ kernel of that layer $\mathbf{W}_i$ (the layer index is removed for the sake of simplicity) with input to the layer $\mathbf{X}$ and adding the bias term $b_i$:

$$\mathbf{Y}_i = \mathbf{W}_i * \mathbf{X} + b_i \tag{4}$$

where $*$ denotes the convolution operator. Considering this, the modulation in (3) can be rewritten as:

$$\hat{\mathbf{Y}}_i = (\eta_i \mathbf{W}_i) * \mathbf{X} + (\eta_i b_i + \gamma_i) \tag{5}$$

We define the modulated kernel, and the modulated bias as $\hat{\mathbf{W}}_i = \eta_i \mathbf{W}_i$, and $\hat{b}_i = \eta_i b_i + \gamma_i$. Note that the modulated kernel is calculated by scaling the whole elements of the original kernel with a single scalar number. Then considering (3)-(5), the modulation in MMAML can be interpreted as convolving the input with this modulated kernel and adding the modulated bias term. The validity of this new interpretation is verified with experimental results (please refer to supplementary material). Based on this new interpretation, the modulated kernels used for different tasks are heavily bound

**Algorithm 2:** Kernel Modulation Algorithm

**Require:** task distribution $p(\mathcal{T})$, learning rate $\alpha$
1  Randomly initialize $\theta, \varphi, \phi$
2  **while** *not done* **do**
3      Sample batch of tasks $\mathcal{T}_i \sim p(\mathcal{T})$
4      **for** *all $\mathcal{T}_i$* **do**
5          Infer $\boldsymbol{v}_{\mathcal{T}_i} = h_\varphi(\mathcal{S}_i)$
6          Generate $M_{\mathcal{T}_i}^{(l)}$ and $\Delta b_{\mathcal{T}_i}^{(l)}$ using (8) and (9)
7          Modulate $\theta$ using (6) and (7) to obtain $\hat{\theta}_{\mathcal{T}_i}$
8          Compute $\mathcal{L}_{\mathcal{T}_i}(\mathcal{Q}_i; \hat{\theta}_{\mathcal{T}_i}, \mathcal{S}_i)$ using $f_{\hat{\theta}_{\mathcal{T}_i}}$
9      **end**
10      Update $\theta \leftarrow \theta - \alpha \nabla_\theta \sum_{\mathcal{T}_i \sim p(\mathcal{T})} \mathcal{L}_{\mathcal{T}_i}(\mathcal{Q}_i; \hat{\theta}_{\mathcal{T}_i}, \mathcal{S}_i)$
11      Update $\varphi \leftarrow \varphi - \alpha \nabla_\varphi \sum_{\mathcal{T}_i \sim p(\mathcal{T})} \mathcal{L}_{\mathcal{T}_i}(\mathcal{Q}_i; \hat{\theta}_{\mathcal{T}_i}, \mathcal{S}_i)$
12      Update $\phi \leftarrow \phi - \alpha \nabla_\phi \sum_{\mathcal{T}_i \sim p(\mathcal{T})} \mathcal{L}_{\mathcal{T}_i}(\mathcal{Q}_i; \hat{\theta}_{\mathcal{T}_i}, \mathcal{S}_i)$
13  **end**

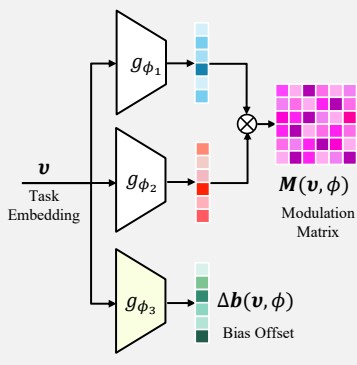

Figure 4: Simplified design for modulation parameter generator.

together by $\mathbf{W}$ and are just scaled versions of it. Since the base kernel is shared across all tasks and scaling gives not many degrees of freedom for each task to produce its desired modulated kernel, different tasks fight for the model capacity, and the negative transfer between tasks from different distributions can lead to training a compromised base kernel.

### 5.2  Kernel Modulation

We propose to modulate the base network by changing the whole parameters within each layer using the modulation parameters conditioned on the task. This gives more capacity for each task to generate its desired kernel. To do this, for every single parameter within the network, a modulation parameter is generated. For a task $\mathcal{T}$, and for each layer $l = 1, \ldots, L$ of the network, let $\mathbf{M}^l(\boldsymbol{v}_\mathcal{T}, \phi)$ denote the modulation matrix generated for modulating the base kernel $\mathbf{W}^l$ of that layer. Among different design choices, intuitively, if we consider the modulated kernels as perturbations around the base kernel, by learning to generate the residual weights, a more powerful base kernel can be learned via enabling proper knowledge transfer across different modes of the task distribution. In our case, this can be achieved by adding the unit matrix $\mathbf{J}$ (all-ones matrix) to the modulation matrix and then perform Hadamard multiplication between the result and the base kernel as follows:

$$\hat{\mathbf{W}}_\mathcal{T}^l = \mathbf{W}^l \odot (\mathbf{J} + \mathbf{M}^l(\boldsymbol{v}_\mathcal{T}, \phi)) \tag{6}$$

For the bias of each layer we simply add an offset term $\Delta \mathbf{b}^l(\boldsymbol{v}_\mathcal{T}, \phi)$:

$$\hat{\mathbf{b}}_\mathcal{T}^l = \mathbf{b}^l + \Delta \mathbf{b}^l(\boldsymbol{v}_\mathcal{T}, \phi) \tag{7}$$

The modulated parameters for task $\mathcal{T}$ are therefore a collection of modulated kernels and biases, i.e., $\hat{\theta}_\mathcal{T} = \{\hat{\mathbf{W}}_\mathcal{T}^1, \ldots, \hat{\mathbf{W}}_\mathcal{T}^L, \hat{\mathbf{b}}_\mathcal{T}^1, \ldots, \hat{\mathbf{b}}_\mathcal{T}^L\}$. The modulation parameters for each task are generated through processing task embeddings of that task $\boldsymbol{v}_\mathcal{T}$ by parameter generator network $\mathbf{g}_\phi$ (Figure 3). Similar to [1], assuming $\mathbf{g}_\phi$ to be MLP, saves lots of computations where a separate parameter generation network $\mathbf{g}_\phi^l$ used for each layer. The main concern with using an MLP is the large number of parameters required in $\mathbf{g}_\phi^l$ to generate the modulation parameters $\Delta \mathbf{b}^l$ and $\mathbf{M}^l$. Because we want to produce the modulation parameters for the whole parameters of a deep neural network, we need to employ some sort of parameter reduction in $\mathbf{g}_\phi$ to prevent an explosion of parameters. Instead of following current complicated algorithms such as pruning the weights of network [44] or redesigning the operations [45], we propose a simple design for $\mathbf{g}_\phi$ which substitutes an MLP module with three smaller ones (Figure 4). In this design, the modulation matrix is generated by processing task embeddings $\boldsymbol{v}_\mathcal{T}$ using two modules, and then performing the outer product operation on the output of these two modules:

$$\mathbf{M}^l(\boldsymbol{v}_\mathcal{T}, \phi) = \mathbf{g}_{\phi_1}^l(\boldsymbol{v}_\mathcal{T}) \otimes \mathbf{g}_{\phi_2}^l(\boldsymbol{v}_\mathcal{T}) \tag{8}$$

The produced matrix is then reshaped to match the kernel shape in that layer. The bias offset of each layer is generated by processing the embedding vector $\boldsymbol{v}_{\mathcal{T}}$ using the third module:

$$\Delta \mathbf{b}^l(\boldsymbol{v}_{\mathcal{T}}, \phi) = \mathbf{g}^l_{\phi_3}(\boldsymbol{v}_{\mathcal{T}}) \tag{9}$$

Our experiments show that compared to a single MLP, the proposed simplified structure decreases the number of parameters in $\mathbf{g}_\phi$ by a factor of **150**, and also provides better generalization performance. A detailed discussion can be found in supplementary. A similar approach to our simplified structure for parameter generator is proposed in [46]. However in [46], it is used to filter out noisy gradients rather than generating task-aware parameter for few-shot learning done in our work. The overall multimodal meta-learning algorithm using the proposed **K**ernel **M**odu**L**ation (KML) scheme is summarized in Algorithm 2. Note that considering the new interpretation presented in (5), our KML can be thought of as a generalization of FiLM. So we can expect that applying KML to the areas improved by FiLM may bring some further improvements. In addition to the few-shot classification results provided in section 6, we have also provided some results on visual reasoning in the supplementary which shows better performance of proposed KML over FiLM.

## 6 Experiments and Analysis

We evaluate the proposed model in both multimodal and unimodal few-shot classification scenarios. Following [1], to create a multimodal few-shot image classification meta-dataset, we combine multiple widely used datasets (Omniglot [47], mini-Imagenet [12], FC100 [48], CUB [49], and Aircraft [50]). Most of our experiments were performed by modifying the code accompanying [1], and for a fair comparison, we follow their experimental protocol unless specified. The details of all datasets, experimental setups, and network structure can be found in the supplementary material. Note that when using an optimization-based meta-learner ($f_\theta$) like MAML, our KML idea can easily be extended to Reinforcement Learning (RL) environments. More details and experimental results on RL can be found in supplementary.

**Multimodal Few-shot Classification Results.** For multimodal few-shot classification, similar to [1], we train and evaluate models on the meta-datasets with two modes (Omniglot and mini-Imagenet), three modes (Omniglot, mini-Imagenet, and FC100), and five modes (all the five datasets). This meta-dataset includes a few-shot classification of characters, natural objects with different statistics and also fine-grained classification of different bird and aircraft types. Due to the large discrepancies between datasets, the possibility of negative transfer increases which makes few-shot classification more challenging. We also combine the mini-ImageNet and FC100 datasets to construct another two-mode meta-dataset. Both mini-ImageNet and FC100 include samples of natural objects and few-shot tasks generated from them probably have a similar underlying structure. However, the few-shot classification of FC-100 tasks is a bit challenging due to the smaller image size. So, here we aim to investigate the possibility of knowledge transfer from mini-ImageNet to FC100. We use two different meta-learners namely ProtoNet [11], and MAML [19], and several baselines considering each meta-learner. The first baseline is training and testing multiple separate meta-learners, one for each dataset. The second baseline is considered when meta-learner has no access to task mode information and simply trained and tested in the combination of tasks from different modes. The third baseline is the MMAML [1] which to the best of our knowledge is the only proposed algorithm for multimodal few-shot learning. A variant of this baseline (MProtoNet) is also used here by replacing the ProtoNet meta-learner with MAML in the general framework of [1].

Multimodal few-shot image classification results are shown in Table 1. Comparing MProtoNet with MMAML, we can see that replacing ProtoNet with MAML considerably improves the classification accuracy. We believe that most of the extra accuracy gained in the multimodal framework by replacing MAML with ProtoNet results from the proper training of modulation network due to improved gradient flow from meta-learner. A detailed experiment can be found in supplementary. Additionally, as the results in Table 1 suggest, the proposed KML scheme considerably improves the performance of multimodal few-shot classification with both meta-learners. The improvement in *5-shot* scenarios is higher because more clues about the task provide more accurate embedding vectors to generate the modulation parameters. We have also included the detailed accuracy for 2Mode$^\dagger$ scenario consisting of mini-ImageNet and FC100 in table 2. This shows on average a positive knowledge transfer from mini-ImageNet to FC100 dataset in terms of increased classification

Table 1: Meta-test accuracies on the multimodal few-shot image classification with 2, 3, and 5 modes. The accuracy values are mean of 1000 randomly generated test tasks, and the $\pm$ shows 95% confidence interval over tasks. 2Mode$^\dagger$ and 2Mode indicate the combination of mini-ImageNet with FC100 and Omniglot, respectively. * Results produced by code provided in [1]. ** Our Implementation.

| Setup | | Method | | | |
|---|---|---|---|---|---|
| | | MAML [19]* | Multi-MAML | MMAML [1]* | MMAML+KML (ours) |
| 2Mode$^\dagger$ | 1-shot | 40.53±68% | 39.27±0.76% | 39.11±0.62% | **40.73±0.66%** |
| | 5-shot | **54.11±0.63%** | 53.51±0.72% | 52.02±0.63% | 53.72±0.60% |
| 2Mode | 1-shot | 65.18±0.61% | 66.77±0.68% | 67.67±0.63% | **68.01±0.59%** |
| | 5-shot | 74.18±0.57% | 73.07±0.61% | 73.52±0.71% | **77.02±0.66%** |
| 3 Mode | 1-shot | 54.40±0.56% | 56.01±0.66% | 57.35±0.61% | **57.68±0.59%** |
| | 5-shot | 66.51±0.54% | 65.92±0.62% | 64.21±0.57% | **67.12±0.55%** |
| 5Mode | 1-shot | 47.19±0.49% | 48.33±0.58% | 49.53±0.50% | **50.31±0.49%** |
| | 5-shot | 58.13±0.48% | 59.20±0.52% | 58.89±0.47% | **60.51±0.47%** |
| | | ProtoNet [11]** | Multi-ProtoNet | MProtoNet [1]** | MProtoNet+KML (ours) |
| 2Mode$^\dagger$ | 1-shot | 43.05±0.58% | 43.42±0.56% | 43.57±0.59% | **44.40±0.65%** |
| | 5-shot | 57.70±0.59% | 56.73±0.64% | 56.03±0.64% | **59.31±0.62%** |
| 2Mode | 1-shot | 69.55±0.54% | 70.17±0.61% | 70.60±0.56% | **73.69±0.52%** |
| | 5-shot | 75.12±0.41% | 75.33±0.46% | 75.72±0.47% | **79.82±0.40%** |
| 3 Mode | 1-shot | 58.14±0.49% | 59.89±0.50% | 59.62±0.54% | **62.08±0.54%** |
| | 5-shot | 66.84±0.44% | 67.03±0.44% | 67.51±0.47% | **70.03±0.43%** |
| 5Mode | 1-shot | 49.31±0.53% | 50.69±0.57% | 51.75±0.52% | **56.72±0.46%** |
| | 5-shot | 58.91±0.51% | 59.88±0.54% | 59.95±0.42% | **64.91±0.38%** |

Table 2: Meta-test accuracies including the performance on each dataset.

| Method | Datasets | | | | | |
|---|---|---|---|---|---|---|
| | mini-ImageNet | | FC100 | | Overall | |
| | 1shot | 5shot | 1shot | 5shot | 1shot | 5shot |
| MAML1 | 41.70±0.91% | **60.03±0.88%** | — | — | 39.27±0.86% | 53.51±0.82% |
| MAML2 | — | — | 36.84±0.81% | 47.08±0.74% | | |
| MAML [19] | **44.11±0.88%** | 59.34±0.78% | 37.01±0.82% | **49.01±0.73%** | 40.53±0.68% | **54.11±0.63%** |
| MMAML [1] | 42.70±0.86% | 56.93±0.78% | 35.50±0.75% | 47.11±0.77% | 39.11±0.72% | 52.02±0.63% |
| MAML+KML(ours) | 44.01±0.82% | 58.95±0.76% | **37.47±0.86%** | 48.45±0.76% | **40.73±0.66%** | 53.72±0.60% |
| ProtoNet1 | 47.19±0.58% | 61.11±0.53% | — | — | 43.42±0.56% | 56.73±0.64% |
| ProtoNet2 | — | — | 39.61±0.54% | 52.35±0.57% | | |
| ProtoNet [11] | 45.62±0.82% | **61.74±0.75%** | 40.36±0.76% | 53.66±0.77% | 43.05±0.58% | 57.70±0.59% |
| MProtoNet [1] | 47.06± 0.80 | 60.56± 0.77 | 40.09± 0.77 | 51.50± 0.83 | 43.57± 0.59 | 56.03± 0.64 |
| MProto+KML(ours) | **48.34± 0.86%** | 61.20± 0.77% | **40.48± 0.77%** | **54.42± 0.76%** | **44.40± 0.65%** | **59.31± 0.62%** |

accuracy compared to a single meta-learner trained and tested on FC100. The improved accuracy of proposed KML over previous modulation comes at the cost of parameter and computational overhead. A detailed discussion can be found in the supplementary. In addition to multimodal scenario, KML also brings considerable improvement in the conventional unimodal scenario (e.g., mini-ImageNet). Please see supplementary for more details and results.

**Transference Results.** In addition to classification results, here we analyse the performance of the proposed KML method on improving knowledge transfer. To this end, we apply the proposed transference metric to analyse the knowledge transfer from 300 randomly sampled mini-ImageNet meta-train tasks to another random meta-test FC100 task. We use Algorithm 1 to calculate the transference during training and the transference histogram is plotted in figure 5 for both ProtoNet and proposed MProtoNet+KML. In each sub-figure, we have included the results for both methods

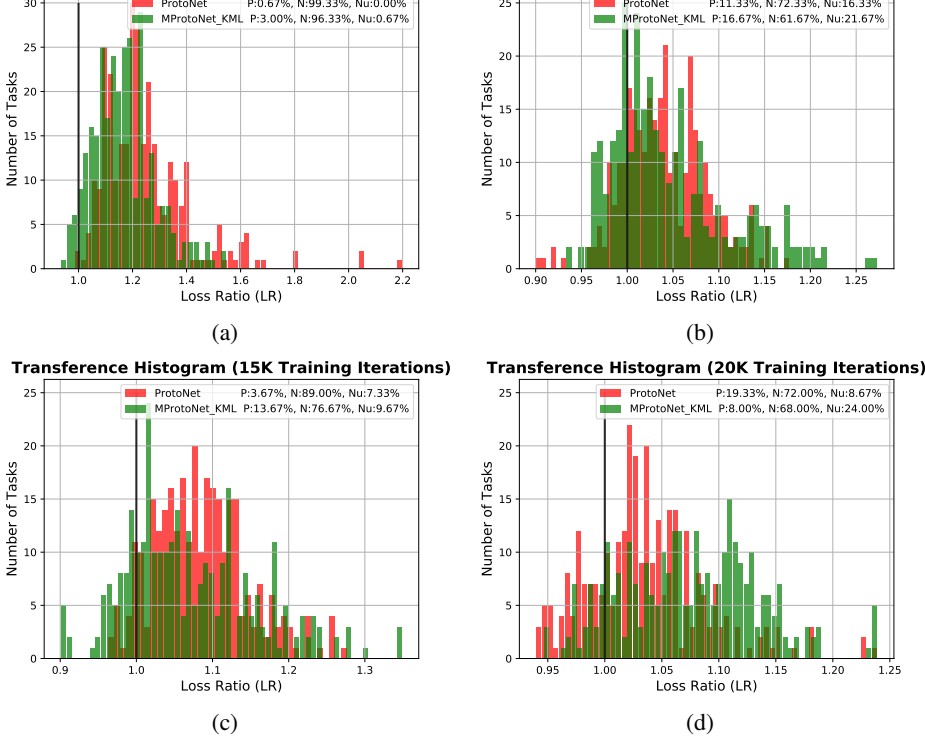

Figure 5: Transference Histogram from mini-ImageNet to FC100 task.

for ease of comparison. A vertical line is also drawn to specify the threshold between positive and negative transference. The percentage of the Positive (P), Negative (N) and Neutral (Nu) transference is also shown for each method. For classifying a source task as neutral, we use an interval around the threshold that the loss change is negligible and corresponds to less than 0.05% change in the meta-test accuracy. Note that in average, proposed KML also performs better in micro-level in terms of increasing positive knowledge transfer and reducing the negative transfer.

The cross mode knowledge transfer can be investigated between different datasets. An interesting behavior that we observed during our analysis is that even Omniglot few-shot tasks –which are considered as easy tasks and can be handled very well with current meta-learning algorithms– can have positive knowledge transfer to challenging FC100 tasks specially in later training iterations. We believe in this case Omniglot tasks act more like a regularizer as they emphasize on simple yet strong features and prevent the model from overfitting to FC100 meta-train task (please find the analysis results in supplementary). Note that knowledge transfer can also be analysed within tasks of a single dataset as in some datasets like mini-ImageNet classes vary so much in few-shot setup (please see supplementary for more details).

## 7 Discussion and Conclusion

In this work, we propose a new quantification method for understanding knowledge transfer in multimodal meta-learning. We then propose a new interpretation of the modulation mechanism in MMAML. These lead us to propose a new multi-modal meta-learner inspired by hard parameter sharing in multi-task learning. Our extensive experimental results show that the proposed method achieves substantial improvement over the best results in multimodal meta-learning. While our major focus is on multimodal meta-learning, our work also attempts to shed light on conventional unimodal meta-learning. Limitations are discussed in the supplementary.

## Acknowledgement

This project was supported by SUTD project PIE-SGP-AI-2018-01. This research was also supported by the National Research Foundation Singapore under its AI Singapore Programme [Award Number: AISG-100E2018-005]. The authors are grateful for the fruitful discussion with Yuval Elovici, and Alexander Binder.

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
