# Revisit Multimodal Meta-Learning
# through the Lens of Multi-Task Learning
# —Supplementary Material—

**Milad Abdollahzadeh, Touba Malekzadeh, Ngai-Man Cheung**
Singapore University of Technology and Design
{milad_abdollahzadeh, touba_malekzadeh, ngaiman_cheung}@sutd.edu.sg

This supplementary material provides results for additional experiments and details to reproduce our results that could not be included in the paper submission due to space limitations.

- **Sec. A** provides additional analysis on our meta-learners based on *hard parameter sharing* idea in MTL. The analysis shows the optimal layer sharing configurations for multimodal and unimodal task distributions. The analysis further justifies the configurations used in the main paper.

- In **Sec. B**, additional transference analysis for both multimodal and unimodal few-shot classification are provided. The additional results are consistent with those in the main paper, and support the finding: *knowledge transfer between different modes happens in the later stages of the training.* Also, results show that *multimodality in terms of transference* also exists in conventional unimodal few-shot learning at a micro-level. This further supports the applicability of proposed KML scheme for conventional few-shot learning.

- **Sec. C** provides the Reinforcement Learning results for proposed KML scheme.

- **Sec. D** provides additional experiments to verify the proposed interpretation of the modulation scheme in MMAML [1]. More specifically, the new interpretation is compared with original Feature-wise Linear Modulation (FiLM) scheme used in MMAML in terms of generated feature maps and also met-test results. This comparison shows same featuremap values for CNN layers and the similar training results in both versions.

- **Sec. E** discusses the parameter reduction of the proposed simplified parameter generator network compared to a single MLP. Results show that for a 4 Layer CNN, proposed simplified structure provides better generalization performance and reduces the number of parameters by a factor of 152.

- **Sec. F** addresses the parameter and computational overhead of the proposed KML compared to original FiLM used in MMAML.

- **Sec. G** provides additional experimental results for higher ranks of the proposed simplified structure.

- In **Sec. H** the results of applying the proposed KML algorithm for visual reasoning is presented.

- **Sec. I** discusses two roles of the meta-learner in the multimodal meta-learner framework: affecting the meta-learning performance, and *feeding the modulation network for gradient update*. Experimental results show that when ProtoNet [2] is replaced by MAML [3] in general framework of MMAML , a more stronger task encoder is learned which can lead to generating more accurate modulation parameters, and consequently improve the generalization performance.

- The details of datasets, network structure and hyperparameters used in all experiments are provided in **Sec. J**.

- Finally, the limitations, broader impact and amount of compute are discussed in **Sec. K**.

35th Conference on Neural Information Processing Systems (NeurIPS 2021).

# A Additional Few-Shot Classification Results

In this section we include additional experimental results for both unimodal and multimodal few-shot classification. Note that here we focus on the analysis of our proposed meta-learner considering the hard parameter sharing concept in MTL. We recall that, by hard parameter sharing for a layer, similar to MTL, we mean the layer that is not modulated using proposed KML and all parameters are shared between different few-shot tasks. For example, in the case of "1$^{st}$ Layer Shared", we mean we do not apply the modulation on the parameters of the first layer (lines 5,6,7,11 and 12 are bypassed for these parameters in Algorithm 2). While for the remaining layers, the modulated parameters are generated and applied in the inner-loop, and then the modulation network is updated in the outer-loop following the procedure in Algorithm 2. So, when all of the layers are shared, the algorithm reduces to the vanilla meta-learner.

**Unimodal Few-Shot Classification.** When compared to a multimodal scenario, in conventional unimodal few-shot setup, there could be less negative knowledge transfer between few-shot tasks (see Sec. B for an example). In this case, from the experiences in the MTL domain, we expect the performance to be increased when some layers are shared between tasks (specially earlier layers which encode low-level features [4]). The meta-test accuracies for different number of shared layers in a 4-layer CNN are shown in table 1. In this case, no shared layers means that all of the layers are modulated using task-aware KML scheme. In contrast, all layers shared means that no modulation is applied. Based on these results, proposed KML improves the unimodal few-shot classification by up to 2.5% compared to the vanilla meta-learner. Additionally, modulating only third and fourth layers of the CNN on average yields the best results. The details of network architecture and hyperparameters used for this experiment can be found in Sec. J.

Table 1: Meta-test accuracies on unimodal few-shot classification for different number of shared layers.

| Shared Layers | mini-ImageNet | | tiered-ImageNet | |
|---|---|---|---|---|
| | 1-shot | 5-shot | 1-shot | 5-shot |
| **No Shared Layers** | 53.18±0.51% | 67.18±0.39% | 54.36± 0.39% | 71.84± 0.27% |
| **1$^{st}$ Layer** | 53.54± 0.66% | **68.07±0.45%** | 54.22± 0.35% | 71.93± 0.28% |
| **1$^{st}$ & 2$^{nd}$ Layers** | **54.10± 0.61%** | 67.31± 0.35% | **54.67± 0.39%** | **72.09± 0.27%** |
| **1$^{st}$, 2$^{nd}$ & 3$^{rd}$ Layers** | 52.83± 0.57% | 66.98± 0.44% | 54.10± 0.37% | 71.68± 0.29% |
| **All Layers (ProtoNet)** | 51.55±0.51% | 65.83±0.36 % | 53.01±0.33% | 70.11±0.29% |

**Multimodal Few-Shot Classification.** In contrast, in multimodal scenario, few-shot tasks are more diverse. So, the negative transfer can happen more often due to the larger discrepancy between the characteristics of the different datasets. Therefore, here we expect the performance to degrade by sharing layers between tasks from different modes. The experimental results for sharing different number of layers in a 4-layer CNN are shown in table 2 for 2Mode multimodal classification (including mini-ImageNet and FC100). Similar results are obtained for other modes. As the results suggest, sharing layers between a diverse set of tasks degrades the performance, and the best results obtained when all of the layers are modulated.

Considering this behavior, for multimodal few-shot classification, the optimal layer sharing configuration is to modulate all layers and use no shared layers. Also for unimodal scenario, sharing first two layers produces the best results due to less negative transfer in unimodal scenario. These results justify the configurations used in the main paper.

# B Additional Experimental Results on Transference Analysis

Here we provide some additional results on transference analysis that could not be included in the main text due to the lack of space. First, we provide some additional results on the transference from mini-ImageNet meta-train tasks to FC100 target tasks. Then, the results for transference analysis from Omniglot to FC100 is discussed. Finally, we provide some results for transference between tasks within a conventional unimodal few-shot learning.

Table 2: Meta-test accuracies on multimodal few-shot classification by including hard parameter sharing.

| Shared Layers | 2Mode$^\dagger$ | |
|---|---|---|
| | 1-shot | 5-shot |
| **No Shared Layers** | **44.40$\pm$0.65%** | **59.31$\pm$0.62%** |
| **1$^{st}$ Layer** | 44.18$\pm$ 0.64% | 59.07$\pm$0.60% |
| **1$^{st}$ & 2$^{nd}$ Layers** | 43.77$\pm$ 0.65% | 58.65$\pm$ 0.59% |
| **1$^{st}$, 2$^{nd}$ & 3$^{rd}$ Layers** | 43.59$\pm$ 0.60% | 58.40$\pm$ 0.61% |
| **All Layers (ProtoNet)** | 43.05$\pm$0.58% | 57.70$\pm$0.59 % |

**Network Structure.** In the transference analysis experiments using ProtoNet and MProtoNet+KML networks, we exactly use the same structure and hyperparameters discussed in Sec. J of this supplementary.

### B.1   Cross-Mode Transference in Multi-Modal Few-Shot Learning

**Transference from miniImageNet to FC100.** Here, we repeat the transference analysis from 300 randomly sampled mini-ImageNet meta-train source tasks into a randomly sampled target tasks from FC100. We train the model with multimodal dataset (including both mini-ImageNet and FC100 tasks). For extracting the transference histogram in an specific epoch, we use the network parameters in this epoch as initial parameters for transference analysis. Then using Algorithm 1 (main paper), first we calculate the loss on target meta-test task using initial parameters. Then, for each source task, we use the data from that task to calculate the adapted parameters to that task and then calculate the loss of target task on adapted parameters. Then, transference from each task is the ratio between the loss of target task after and before adapting.

This target classification task contains samples from "otter, girl, dolphin, raccoon, and skunk". The classes are disjoint enough and the sampled data points are clean which makes classification task a less challenging one to handle. On the other hand, most of the classes share similar underlying structure as in the animal subcategories of the mini-ImageNet tasks. So, as we expect, the ProtoNet also performs better in meta-target task with the information provided by the source mini-ImageNet tasks. The transference histograms during training are shown in figure 1. Based on the transference results, following points can be considered:

- In the beginning iterations of the training, most of the source tasks from mini-ImageNet have negative transference on the target task. The probable reason could be that at the beginning stages, the model has not learned to generalize and still tries to remember the meta-training tasks and corresponding samples (Please note that the LR is measured on a meta-test task which consists of unseen classes under the few-shot problem setting).
- As training proceeds, the percentage of the positive transference increases. After learning the useful features, the network begins to overfit to training tasks and as a result its generalization performance (knowledge transfer to FC100 meta-test task) degrades.

Overall, our additional results are consistent with those in the main paper, and support the observation: **In the case of cross mode knowledge transfer, negative transference occurs at the beginning iterations and increasingly more positive transference occurs as training proceeds.** As the major advantage of cross mode knowledge transfer is to improve generalization to better handle unseen test tasks, it is reasonable to observe positive transference in the later iterations when the networks learn features for generalization.

**Average Transference.** The average transference values from 300 meta-train mini-ImageNet tasks to 100 meta-test FC100 tasks are shown in figure 2. For calculating each point in this figure, we have averaged the transference results from 300 source tasks to each task and then computed the average along all target tasks in that training iteration. This figure shows that in the beginning of the training, the information provided by the mini-ImageNet are almost negative for generalization performance on the FC100 meta-test tasks. As training proceeds and the network learns necessary features (and

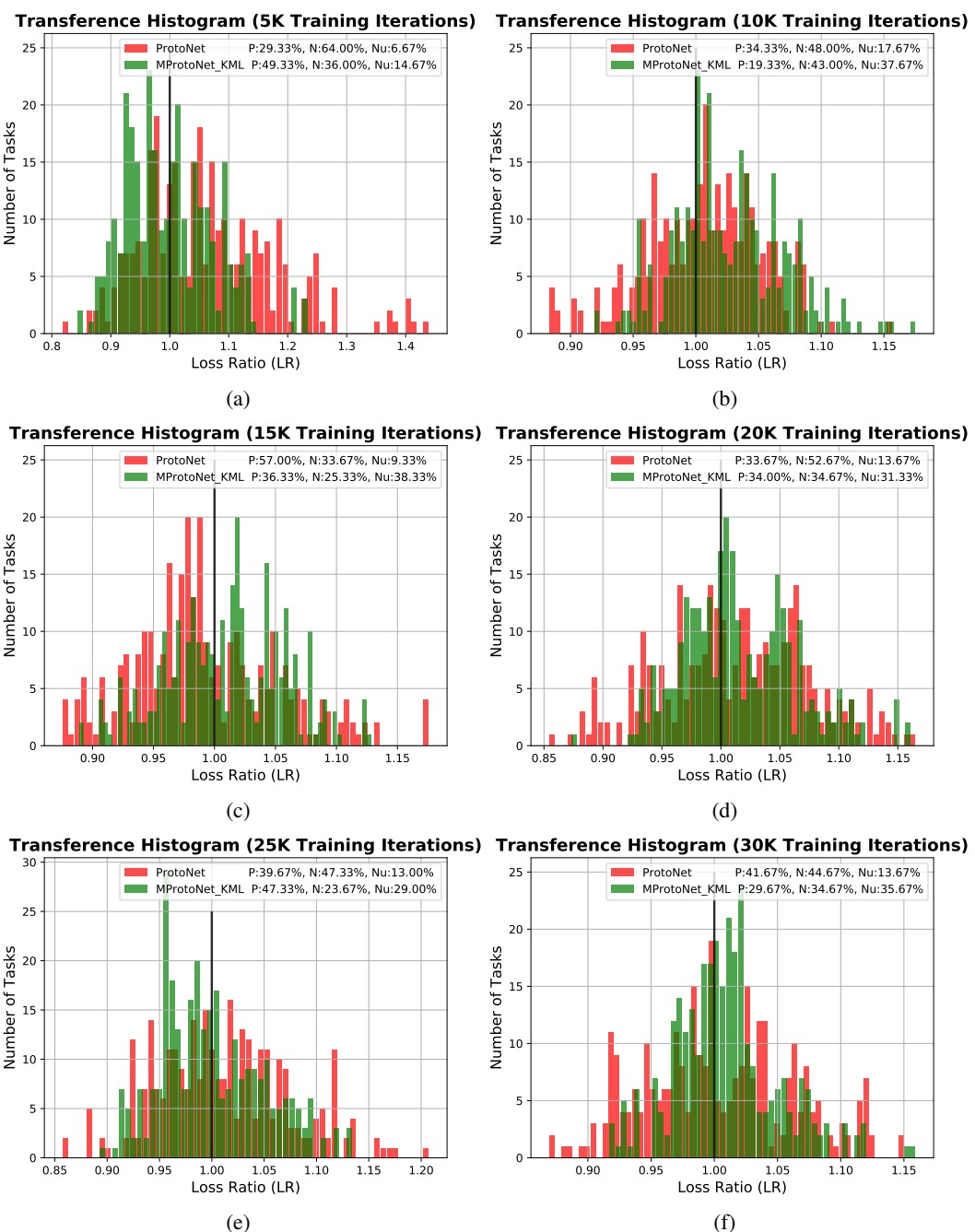

Figure 1: Transference Histogram from mini-ImageNet to FC100 task.

probably low and mid-level ones), it can make good use of additional information provided by the external mini-ImageNet dataset. This means that *cross-mode knowledge transfer occurs on the later iterations of training*. Results also show that on average the proposed method performs better in terms of obtaining the information provided by the source meta-train tasks.

**Transference from Omniglot to FC100.** Here we investigate the transference from Omniglot meta-train tasks to FC100 meta-test target task. Omniglot is a set of black-and-white images of handwritten characters with clean background. Therefore, classification of these tasks requires not much complicated feature representation, and the major challenge is limited number of data samples

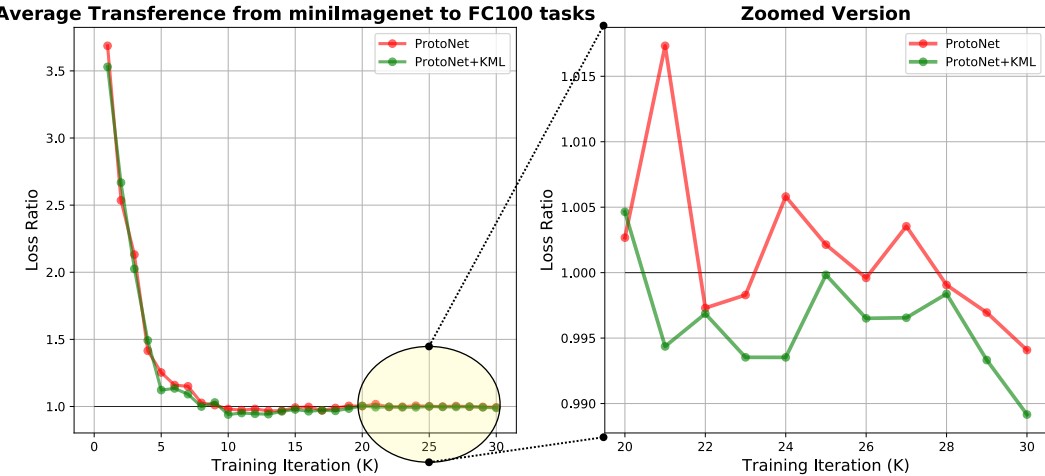

Figure 2: Average transference from 300 mini-ImageNet task to 100 meta-test FC100 tasks.

in each task. Please note that current meta-learning algorithms can easily handle the Omniglot few-shot classification tasks, and their performance is almost saturated on this dataset. However, by including the Omniglot in our analysis, we aim to investigate the dynamics of knowledge transfer from Omniglot to challenging FC100 few-shot tasks.

In this experiment we train both ProtoNet and proposed MProtoNet+KML on a meta-dataset constructed by combining Omniglot and FC100 few-shot tasks. We samples 300 meta-train Omniglot tasks, and a meta-test FC100 task as target task to perform transference analysis. The transference results are shown in figure 3. Considering these results, similarly, as training proceeds the negative knowledge transfer reduces. An interesting phenomena is the high rate of positive transfer in the later training stages. In the later training stages, probably the model overfitts to meta-train classes by learning some features that can not generalize well for meta-test tasks. On the other hand, source meta-train tasks from Omniglot require strong but simpler features due to their samples type. So, a potential reason is that since Omniglot meta-train tasks emphasize on these features, they can prevent overfitting and have a positive impact on the generalization performance on the FC100 meta-test task.

## B.2 Transference within the same mode in Unimodal Few-Shot Learning

In this section, we analyze the knowledge transfer between tasks within a conventional unimodal dataset. While a single dataset is defined to be one mode following the definition in [1], tasks from different classes can have a negative or positive impact on each other within a dataset. To investigate this, we analyze the transference from a number of meta-train miniImageNet tasks to a meta-test task from the same dataset. The experimental setup is exactly like the previous experiments, but the only difference is that we just train the model on the miniImageNet dataset (not combination of datasets). The transference results from 300 randomly sampled miniImageNet meta-train tasks on a target miniImageNet meta-test task is shown in figure 4 for different training iterations.

Transference histograms show that there is a multimodality in terms of transference from meta-train tasks of a dataset to a target meta-test task from the same dataset. This means that a group of tasks have positive knowledge and others have negative. This can be interpreted with our previous findings. For example, in the simplest form, if the target meta-test task includes samples from animal classes, we expect the meta-train tasks from animal classes to have positive knowledge transfer, and tasks from non-animal classes to have negative transfer. Considering this behaviour within tasks in mini-ImageNet dataset, applying proposed KML scheme boosts the performance by reducing negative transfer (also shown in histograms) through assigning task-aware layers.

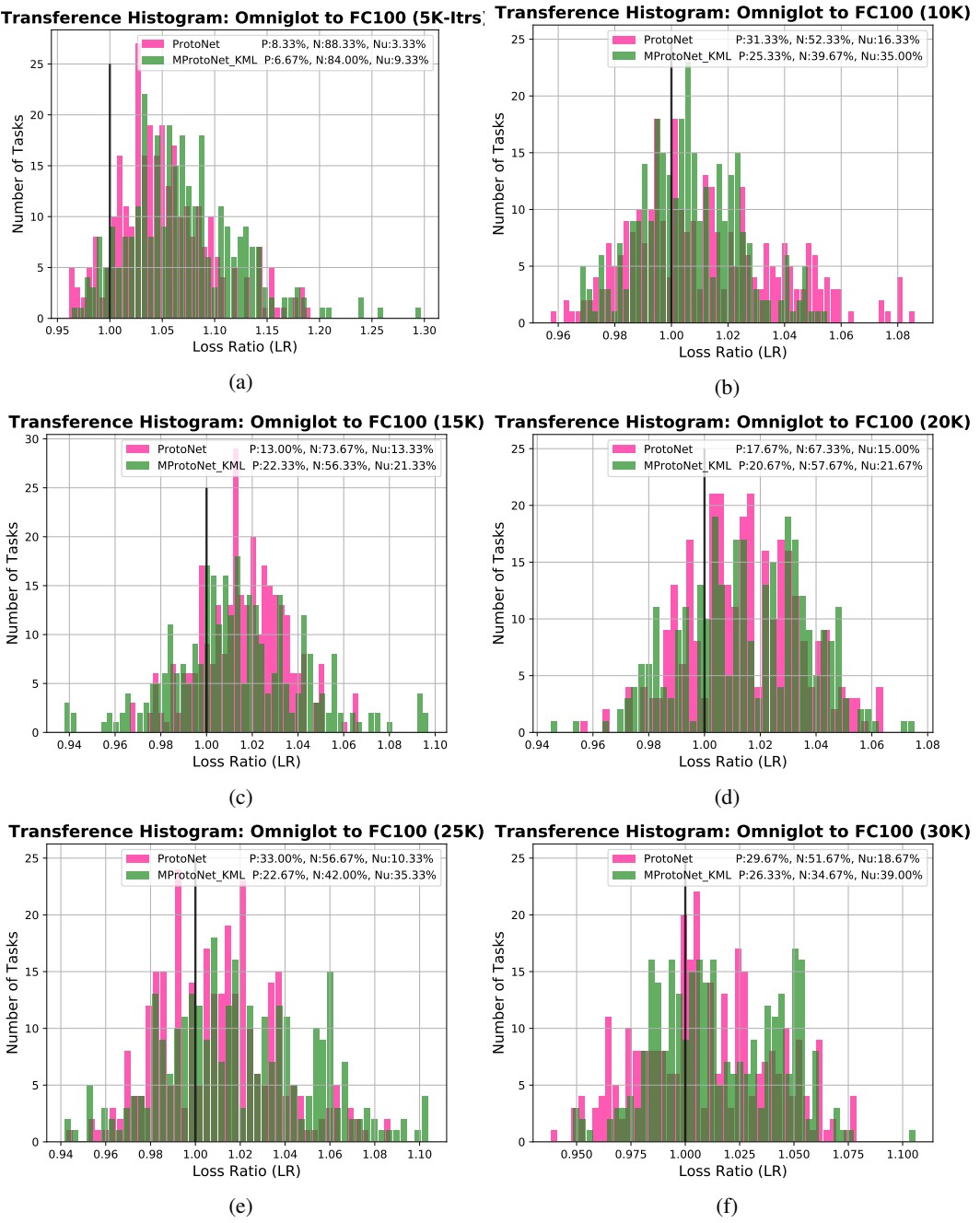

Figure 3: Transference Histogram from Omniglot to FC100 task.

## C Reinforcement Learning Results

The proposed KML idea can be extended to Reinforcement Learning (LR) environments, when using an optimization-based meta-learner like MAML. we have applied our KML algorithm on the official code of MMAML for RL experiments on three different environments used in [1]: Point Mass, Reacher, and Ant. Similar to [1], for each environment, the goals are sampled from a multimodal goal distribution, with similar environment-specific parameters as [1]. To have a fair comparison, we have kept all other hyperparameters the same as the [1]. The mean and standard deviation of cumulative reward per episode for multimodal reinforcement learning problems with 2, 4 and 6

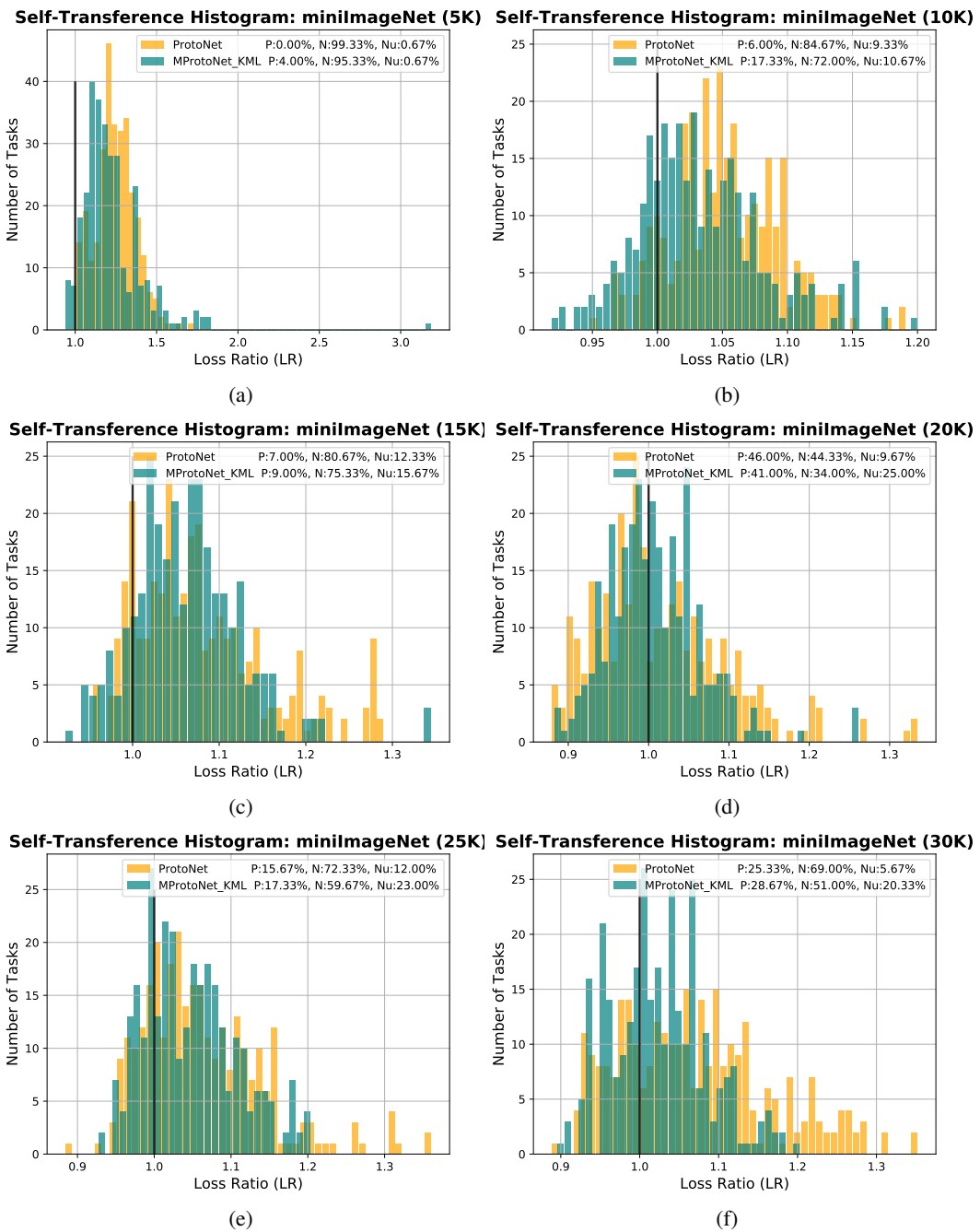

Figure 4: Self-Transference Histogram within mini-ImageNet dataset.

modes are shown in table 3. Results show that our proposed KML can achieve gain over [1] in all of the RL experiment setups.

# D   Verification of New Interpretation

In Sec. 5.1. of the main paper, we propose a new interpretation of modulation scheme used in MMAML. Briefly, we show that feature-wise linear modulation (FiLM) [5] applied to each channel of feature map in MMAML, can be considered as convolving the input with modulated kernel

Table 3: The mean and standard deviation of cumulative reward per episode for multimodal reinforcement learning problems with 2, 4 and 6 modes reported across 3 random seeds.

| Method | Point Mass 2D | | | Reacher | | | Ant | |
|---|---|---|---|---|---|---|---|---|
| | 2Modes | 4Modes | 6Modes | 2Modes | 4Modes | 6Modes | 2Modes | 4Modes |
| MMAML | -136±8 | -209±32 | -169±48 | -10.0±1.0 | -11.0±0.8 | -10.9±1.1 | -711±25 | -904±37 |
| MMAML+KML(ours) | -121±9 | -197±30 | -161±41 | -9.6±1.0 | -10.6±0.7 | -10.6±1.0 | -689±23 | -891±36 |

$\hat{\mathbf{W}}_i = \eta_i \mathbf{W}_i$ and adding the modulated bias term $\hat{b}_i = \eta_i b_i + \gamma_i$. In addition to the mathematical formulation provided in the main paper, we also verify this new interpretation with experiments.

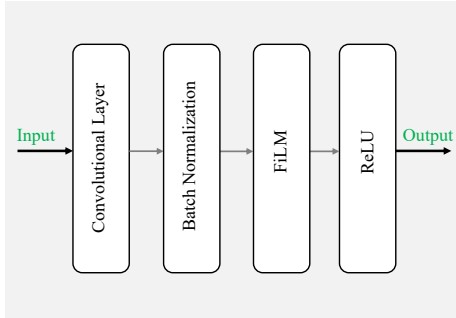

Figure 5: The structure of each layer in MMAML.

*First,*, we compare generated featuremaps by the proposed new interpretation for MMAML (*new*-MMAML) with the ones generated by the original implementation of MMAML. For implementing the new interpretation of MMAML, we use the official code provided by authors with exactly same hyperparameters. Figure 5 shows the structure of each Convolutional Block used in MMAML for few-shot image classification. Original MMAML implementation uses the Batch Normalization (BN) layer before modulation. So, for a fair comparison, we disable the BN layer in MMAML and extract the feature maps after applying FiLM. Then for *new*-MMAML, we simply perform convolution with modulated parameters $\hat{\mathbf{W}}_i$ and $\hat{b}_i$ to produce featuremaps. The produced featuremaps are same for all CNN layers. For example, for a mini-ImageNet classification task, the average error between two produced featuremaps in first layer is around $\mathbf{3.7}\mathrm{e}{-\mathbf{3}}$ while the average absolute value of feature maps is around $\mathbf{1.4}\mathrm{e}{+\mathbf{2}}$. Note that this minor error probably stems from the floating-point round-off error in PyTorch [6].

Table 4: Comparison of meta-test accuracies for original implementation of MMAML with proposed new interpretation (*new*-MMAML) for few-shot classification on multimodal scenario.

| Method | 2 Mode | | 3 Mode | | 5 Mode | |
|---|---|---|---|---|---|---|
| | 1-shot | 5-shot | 1-shot | 5-shot | 1-shot | 5-shot |
| MMAML | 67.67±0.63% | 73.52±0.71% | 57.35±0.61% | 64.21±0.57% | 49.53±0.50% | 58.89±0.47% |
| *new*-MMAML | 67.43±0.61% | 73.64±0.66% | 57.44±0.60% | 64.09±0.61% | 49.23±0.51% | 58.71±0.44% |

*Second*, we compare the training performance of MMAML and *new*-MMAML. In the official implementation of MMAML, the affine transform of the BN layer [7] is disabled, due to the similar functionality performed by FiLM. However, since in *new*-MMAML we are using modulated parameters, we enable the affine transform of the BN layer. Meta-test results are shown in table 4 for different multimodal image classification modes. As the results suggest the *new*-MMAML has almost the same performance as the MMAML which also verifies the proposed interpretation through experiments. The minor difference between the results is due to the difference between the BN layer in two implementations (as discussed).

# E  Parameter Reduction in Proposed Parameter Generator

In Sec. 5.2 of the main paper we have proposed a structure to reduce the number of parameters in the modulation parameter generator network $\mathbf{g}_\phi$. As discussed, proposed structure includes three smaller MLP modules. Here a more detailed comparison between the proposed simplified structure and a single MLP is provided. A standard convolutional layer is parameterized by convolution kernel of size $N_k \times N_k \times N_i \times N_o$ and a bias term of size $N_o$, where $N_k$ is the spatial dimension of kernel, $N_i$ is the number of input channels and $N_o$ is the number of output channels. Then the required number of parameters for an MLP with single hidden layer that takes the task embeddings with size $N_v$ to produce the whole elements for this layer is: $N_v \times (N_k \times N_k \times N_i \times N_o + N_o)$. Instead using the proposed structured MLP, we use three smaller MLPs to produce $N_o$, $N_i \times N_k \times N_k$ and $N_o$ parameters, respectively. Then the parameter reduction ratio compared to single MLP is:

$$\frac{N_k \times N_k \times N_i \times N_o + N_o}{N_k \times N_k \times N_i + 2N_o}$$

Following the structure proposed in [1], the base network consists of four convolutional layer with the channel size 32, 64, 128 and 256. In our KML scheme we intend to produce a modulation parameter for each parameter of the base network using task embedding $v$ (with a dimension of 128) as input. Table 5 compares the number of parameters in single MLP and proposed structure when used as parameter generation network for each layer. The total number is also provided. As one can see, proposed structure reduces the number of the parameters by a factor of **152**.

Table 5: Number of parameters required in each structure as modulation parameter generator $\mathbf{g}_\phi$.

| Layer | Number of parameters | |
|---|---|---|
| | **MLP** | **Proposed Structure** |
| #1 | 114,688 | 11,648 |
| #2 | 2,367,488 | 45,056 |
| #3 | 9,453,568 | 90,112 |
| #4 | 37,781,504 | 180,224 |
| **Total** | **49,717,248** | **327,040** |

We also compare using MLP with proposed structure in terms of convergence speed. We use the same hyperparameters for training both models. The accuracy of meta-validation set during meta-training on 3Mode, 5-way 1-shot setting is plotted in figure 6. We can clearly see that using the proposed structure as modulation parameter generator, the network converges faster and also yields better results in term of accuracy compared to single MLP. We have also provided the meta-test accuracy results for 3Mode few-shot classification in Table 6. These results also support the improved performance of the proposed simplified structure versus single MLP.

Table 6: 2Mode Meta-test accuracy of the proposed simplified structure versus single MLP for 2Mode 5-way scenario.

| Method | Setup | |
|---|---|---|
| | **1shot** | **5shot** |
| **MLP** | 61.22±0.56% | 69.38±0.48% |
| **Proposed Simplified Structure** | **62.08±0.54%** | **70.03±0.43%** |

# F  KML vs FiLM: Parameter and Computational Overhead

Previously, we have demonstrated that replacing FiLM with the proposed KML method significantly improves the accuracy of the meta-learner in both multimodal and conventional unimodal few-shot

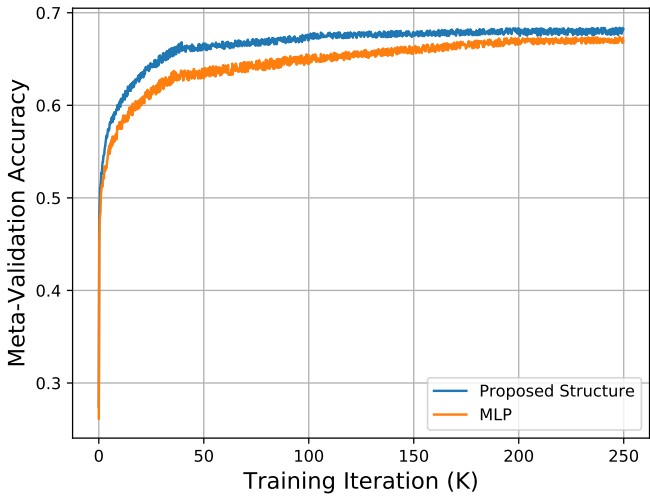

Figure 6: The meta-validation accuracy during meta-training.

classification. KML achieves this substantial improvement by modulating the whole elements of the kernel instead of applying the affine transform on the feature maps (FiLM in [1]). This is done by generating a larger number of parameters compared to FiLM. Here we analyze the overhead introduced by replacing FiLM (existing method in [1]) with KML (our proposed method).

First, we discuss the number of additional parameters introduced by KML. Since the only difference between the two methods is on the generator, we consider this module for comparison. Recalling from section D of the supplementary, the number of parameters required in proposed simplified structure in KML (for a layer) is $N_v \times (N_k \times N_k \times N_i + 2N_o)$. Since in FiLM, only two parameters are generated for each channel of the convolutional layer, the number of parameters in the generator is $N_v \times (2N_o)$. Therefore, the additional overhead of KML for the generator becomes $N_v \times N_k \times N_k \times N_i$ for each layer. Considering we have four convolutional layers in our structure and for each layer, a separate generator is used, in total, KML adds around 261.5 K parameters to the ones in FiLM. Considering this number, for example, the total number of parameters in MProtoNet+KML (our method) increases by 22.9% compared to MProtoNet (existing method in [1]).

Table 7: Training time for MProtoNet and MProtoNet+KML (proposed method) for 2Mode setup.

| Method | Training Time | Accuracy |
|---|---|---|
| **MProtoNet** | 173 minutes | 56.03±0.64% |
| **MProtoNet+KML(proposed)** | 182 minutes | 59.31±0.62% |

Second, in terms of computational overhead, the table 7 shows the total training time for MProtoNet (existing method in [1]) and MProtoNet+KML (ours) for 2Mode (combination of Omniglot and miniImageNet), 5-way 5-shot scenario. As the results show, the computational overhead of the proposed method in training time is around 5.2%. Similar training results are obtained for the other setups (3 Mode, 5Mode). Also please note that the inference time of our method and existing method [1] are almost the same (on average 0.087 seconds for each mini-batch of few-shot tasks).

## G    Simplified Parameter Generator as Low-Rank Approximation

The proposed simplified structure (figure 4 of the main paper) can be considered as a low-rank approximation (1-rank). In the section E of this supplementary, we have shown that this 1-rank approximation achieves better meta-test results compared to a full-rank version. Here we check the results for higher ranks (2-rank and 3-rank). For producing the 2-rank approximation, we produce two different matrices: $M_1$ and $M_2$ using a similar method as (8) in our paper, and then add these two

matrices to generate the final modulation matrix $M = M_1 + M_2$. Please note that this time instead of 3 modules, we have 5 modules in our simplified structure. Two pairs of modules are used to generate the $M_1$ and $M_2$, and the fifth one is used to generate the bias term. We have checked these vectors to be independent. A similar procedure is used to design a 3-rank approximation of the MLP using three different pairs.

Table 8: Meta-test accuracies for 2Mode setup with different rank approximation in simplified parameter generator $g_\phi$.

| Setup | MProtoNet | MProtoNet+KML(1-rank) | MProtoNet+KML(2-rank) | MProtoNet+KML(3-rank) |
|---|---|---|---|---|
| **5way-1shot** | 70.60±0.56% | 73.69±0.52% | 72.12±0.54% | 72.06±0.52% |
| **5way-5shot** | 75.72±0.47% | 79.82±0.40% | 78.94±0.43% | 78.70±0.46% |

The meta-test results for 2Mode classification are shown in table 8. As results suggest, the 2-rank and 3-rank approximations still have better performance compared to the MProtoNet. However, the performance is degraded compared to the 1-rank approximation. The possible reason could be overfitting of 2-rank and 3-rank versions due to more parameters.

## H  KML for Visual Reasoning

We remark that based on the proposed new interpretation of the FiLM scheme, our proposed KML can be seen as a generalization of FiLM. So we can expect that applying KML to the areas improved by FiLM may bring some further improvement. We also declare that the amount of improvement depends on the underlying structure of learning tasks. For example in the case of few-shot learning (especially multimodal distribution), since there could be a significant difference between different tasks (e.g., digit classification vs natural object classification), KML brings a large improvement over FiLM by letting the more powerful adaption of kernels for each few-shot task. Intuitively, this improvement may be less for the applications where more similar kernels are required for different tasks, e.g., visual reasoning on CLEVR dataset where there is a significantly lower variation on image statistics compared to our multimodal few-shot distribution, and the main difference originates from the question, and probably program signal.

Table 9: Results of applying the proposed KML on visual reasoning dataset CLEVR.

| Method | Count | Exist | Compare Numbers | Query Attribute | Compare Attribute | Average |
|---|---|---|---|---|---|---|
| CNN+GRU+FiLM [5] | 94.3% | 99.1% | 96.8% | 99.1% | **99.1%** | 97.7% |
| CNN+GRU+KML(ours) | **96.1%** | **99.5%** | **97.1%** | **99.3%** | **99.1%** | **98.2%** |

We applied the KML to the CLEVR dataset by replacing KML with the FiLM in the official code of the FiLM paper [5]. The results are shown in table 9. As the results show, KML on average improves the FiLM by 0.5% in the 5 question types. Note that KML obtains this improvement while FiLM has achieved very high accuracy already.

## I  Gradient Flow From Meta-learner to Modulation Network

In general multimodal meta-learning framework, meta-learning algorithm plays two important roles. *First*, its meta-learning capability directly affects the multimodal meta-learning performance. *Second*, modulation network is fed with the gradient propagated from the meta-learner in each training iteration. In the case of good gradient propagation, the modulation network can be trained well to predict the task mode and generate powerful modulation parameters. So, good gradient flow from meta-learner to modulation network is as important as the good performance of the meta-learner itself.

**Limitation of MMAML in Terms of Gradient Flow.**  While proposing a generic framework, another limitation of MMAML is due to using MAML as meta-learner. MAML is a simple and elegant meta-learning algorithm, however, backpropagating the gradients through inner-loop (with multiple

updates) requires the second order gradients for optimization. This increases the computational complexity of meta-learning and makes it difficult for gradients to propagate [8]. This problem gets even worse when using MAML in multimodal meta-learning framework where the modulation network and meta-learner are supposed to be trained together in an end-to-end fashion. We empirically found that multimodal meta-learning framework can benefit from using a more simpler meta-learner like ProtoNet [2] which has better gradient propagation due to replacing the inner-loop adaptation with prototype construction.

**Experiment.** To gain a better understanding on the effective modulation network training, we randomly sample 1000 5-mode, 5-way 1-shot meta-test tasks and calculate the task embeddings $v$ for each task using both MMAML and MProtoNet. Then we employ the t-SNE [9] to visualize $v$ in Figure 7. As one can see, in the task embeddings produced by MProtoNet, the embeddings for different modes are separated better.

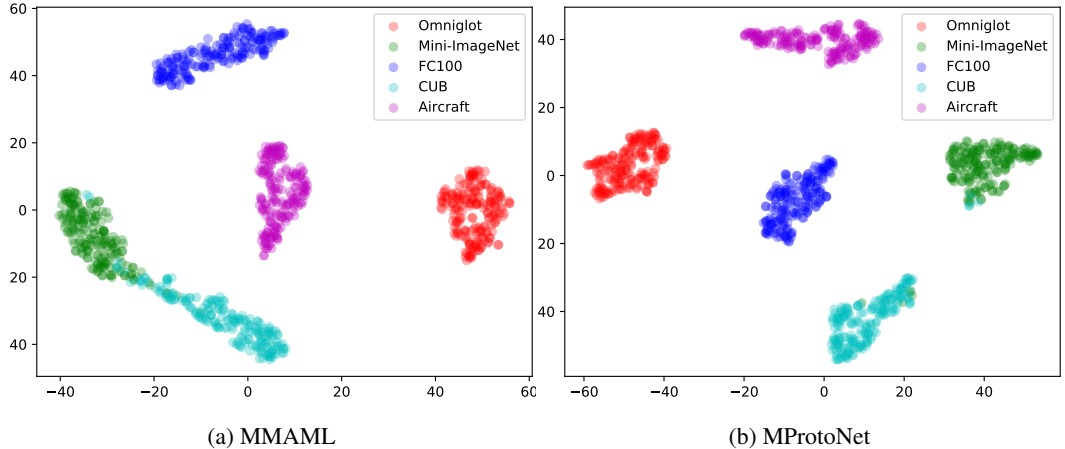

(a) MMAML

(b) MProtoNet

Figure 7: t-SNE plots for task embedding vectors $v$ produced for 1000 randomly generated test tasks in 5Mode, 5-way, 1-shot setup by MMAML and MProtoNet.

## J  Experimental Details

### J.1  Meta-Dataset

To create a meta-dataset for multi-modal few-shot classification, we utilize five popular datasets: OMNIGLOT, MINI-IMAGENET, FC100, CUB, and AIRCRAFT. The detailed information of all the datasets are summarized in Table 10. To fit the images from all the datasets to a model, we resize all the images to 84 × 84. The images randomly sampled from all the datasets are shown in Figure 8, demonstrating a diverse set of modes.

Table 10: Details of Datasets.

| Dataset | Train Classes | Validation Classes | Test Classes | Image Size | Image Channel | Image Content |
|---|---|---|---|---|---|---|
| Omniglot | 4112 | 688 | 1692 | 28×28 | 1 | handwritten digits |
| mini-ImageNet | 64 | 16 | 20 | 84×84 | 3 | natural objects |
| FC100 | 64 | 16 | 20 | 32×32 | 3 | natural objects |
| CUB | 140 | 30 | 30 | ~500×500 | 3 | species of birds |
| Aircraft | 70 | 15 | 15 | ~ 1-2 Mpixels | 3 | types of aircrafts |

### J.2  Meta-Learning Algorithms in Multimodal Framework

Here we provide more details about the meta-learners used in experiments including: MAML [3] and ProtoNet [2].

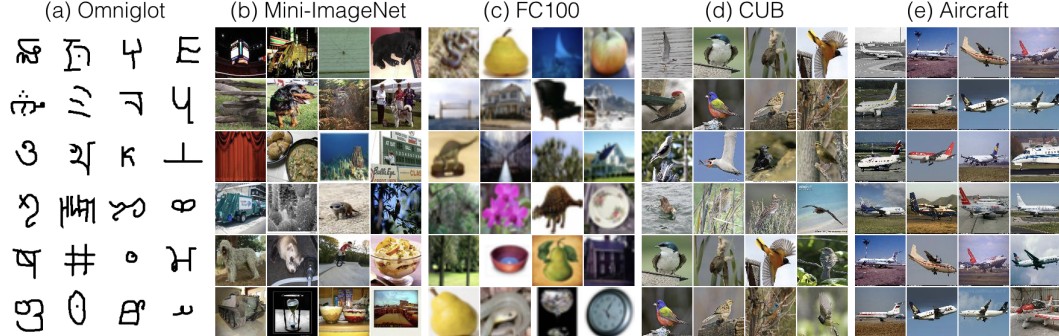

| (a) Omniglot | (b) Mini-ImageNet | (c) FC100 | (d) CUB | (e) Aircraft |

Figure 8: Examples of images used to create multimodal meta-dataset.

**MAML.** MMAML uses the model-agnostic meta-learning (MAML) algorithm [3] as meta-learner in base network. Given a network $f_\theta$, MAML aims to learn a common initialization for weights $\theta$ under a certain task distribution such that it can adapt to new unseen task with a few steps of gradient descent. In multimodal scenario of MMAML, for each task $\mathcal{T}_i$, first the modulation parameters $\boldsymbol{\tau}_i$ are generated. Then base network uses these parameters together with the samples from support set $\mathcal{S}_i$ to adapt the network weights to that task using gradient descent:

$$\theta_i' = \theta - \alpha \nabla_\theta \mathcal{L}_i(f_\theta(\mathcal{S}_i; \boldsymbol{\tau}_i))$$

Where $\mathcal{L}_i(f_\theta(\mathcal{S}_i; \boldsymbol{\tau}_i))$ denotes the loss function on the support set of task $\mathcal{T}_i$. Then the adapted model is evaluated on the query samples of the same task $\mathcal{Q}_i$ which provides the feedback in the form of loss of gradients for generalization performance on that task. The feedback from a batch of tasks is used to update the base network weights $\theta$ to achieve better generalization:

$$\theta \leftarrow \theta - \beta \nabla_\theta \sum_{\mathcal{T}_i} \mathcal{L}_i(f_{\theta_i'}(\mathcal{Q}_i))$$

The modulation network parameters $(\phi, \varphi)$ are also updated in a same manner using the feedback from the generalization performance of the adapted model.

**ProtoNet.** As another multimodal meta-learner, we replace the ProtoNet with MAML due to its ease of training and better gradient flow. In this algorithm, first modulation parameters are generated by processing task samples. Then network parameters are modulated using these parameters to produce modulated parameters $\hat{\theta}_\mathcal{T}$ [1]. ProtoNet computes a *prototype* for each class through an embedding function $f_{\hat{\theta}_\mathcal{T}} : \mathbb{R}^D \to \mathbb{R}^F$ which maps an input sample $\mathbf{x} \in \mathbb{R}^D$ to an $F$-dimensional feature space. The prototype $\mathbf{c}_n$ of a class $n = 1, \dots, N$, is the mean vector of the embedded support samples belonging to that class:

$$\mathbf{c}_n = \frac{1}{|\mathcal{S}_\mathcal{T}^n|} \sum_{x \in \mathcal{S}_\mathcal{T}^n} f_{\hat{\theta}_\mathcal{T}}(\mathbf{x})$$

After generating prototypes, it uses a distance function to produce the distribution over classes for each query sample $\tilde{\mathbf{x}}$ as follows:

$$p_n(\tilde{\mathbf{x}}) = \frac{exp(-d(f_{\hat{\theta}_\mathcal{T}}(\tilde{\mathbf{x}}), \mathbf{c}_n))}{\sum_{n'} exp(-d(f_{\hat{\theta}_\mathcal{T}}(\tilde{\mathbf{x}}), \mathbf{c}_{n'}'))}$$

This predicted distribution is compared with class labels to compute the loss. Then this loss optimized by SGD with respect to network parameters $(\theta, \phi, \varphi)$.

---

[1]Note that, while in our KML algorithm these parameters are produced explicitly, in MProtoNet (variant of MMAML produced in this paper) these are implicitly generated by applying FiLM [5] on each featuremap.

### J.3 Network Structures

#### J.3.1 Base Network

**Multimodal Few-shot Classification Experiments.** In multimodal experiments, for the base network (as meta-learner), we use the exactly same architecture as the MMAML convolutional network proposed in [1]. It consists of four convolutional layers with the channel size 32, 64, 128, and 256, respectively. All the convolutional layers have a kernel size of 3 and stride of 2. A batch normalization layer follows each convolutional layer, followed by ReLU. With the input tensor size of $(n \cdot k) \times 84 \times 84 \times 3$ for an *n-way, k-shot* task, the output feature maps after the final convolutional layer have a size of $(n \cdot k) \times 6 \times 6 \times 256$. For ProtoNet-based architectures, these featuremaps are directly used for constructing prototypes and performing classification. For MAML-based architectures, the featuremaps are average pooled along spatial dimensions, resulting feature vectors with a size of $(n \cdot k) \times 256$. In this case, a linear fully-connected layer takes the feature vector as input, and produce a classification prediction with a size of *n* for *n-way* classification task.

**Unimodal Few-Shot Classification Experiments.** In conventional unimodal few-shot classification, we use the more standard architecture which is slightly different from the one used in MMAML for multimodal scenario. Here, for ProtoNet-based experiments, following the original implementation of the ProtoNet [2], 4 similar convolutional blocks are used. Each block comprises a 64-filter 3×3 convolution, batch normalization layer, a ReLU nonlinearity and a 2×2 max-pooling layer. As also discussed in [1], the slight difference between the multimodal results and unimodal ones reported in previous works is due to the difference in network structure and hyperparameters.

#### J.3.2 Modulation Network

The modulation network includes a task encoder network $h_\varphi$ and a modulation parameter generator network $g_\phi$.

**Task Encoder.** Similar to [1], for the task encoder, we use the exactly same architecture as the base network. It consists of four convolutional layers with the channel size 32, 64, 128, and 256, respectively. All the convolutional layers have a kernel size of 3, stride of 2, and use valid padding. A batch normalization layer follows each convolutional layer, followed by ReLU. With the input tensor size of $(n \cdot k) \times 84 \times 84 \times 3$ for a n-way k-shot task, the output feature maps after the final convolutional layer have a size of $(n \cdot k) \times 6 \times 6 \times 256$. The feature maps are then average pooled along spatial dimensions, resulting feature vectors with a size of $(n \cdot k) \times 256$. To produce an aggregated embedding vector from all the feature vectors representing all samples, we perform an average pooling, resulting a feature vector with a size of 256. Finally, a fully-connected layer followed by ReLU takes the feature vector as input, and produce a task embedding vector $\upsilon$ with a size of 128.

**Modulation Parameter Generator.** Modulation parameter generator structure varies based on the multimodal algorithm. For MMAML and MProtoNet, we follow the design in [1]. In this design, the modulating each channel requires producing two parameters ($\eta_i$ for scaling and $\gamma_i$ for shifting featuremap). Considering the channel size 32, 64, 128 and 256 in base network, four linear fully-connected layers are used to convert task embedding vector $\upsilon$ (with a size of 128) to required modulation parameters. The size of these layers are as follows: $128 \times 64$, $128 \times 128$, $128 \times 256$ and $128 \times 512$ [2]. For KML-based meta-learners (MMAML+KML and MProtoNet+KML), we produce a modulation number for each parameter in the network using the proposed simplified structure. The details explanation on the number of required parameters are discussed in Sec. E of this supplementary material.

#### J.3.3 Hyperparameters

The hyperparameters for all the experiments are shown in Table 11. For comparing our algorithm with previous work, we use exactly the same hyperparameters. We use 15 examples per class for evaluating the post-update meta-gradient for all the experiments, following [3, 10, 1, 2]. In the training of all networks, we use the Adam optimizer with default hyperparameters.

---

[2]For unimodal experiments, these numbers are changed based on the number of filters in the base network.

Table 11: Hyperparameters used in the experiments. $^\dagger$ Halve every 10K Iterations.

| Method | DataSet group | Inner lr | Outer lr | Meta batch-size | Number of updates | Training Iterations |
|---|---|---|---|---|---|---|
| MMAML | – | 0.05 | 0.001 | 10 | 5 | 60000 |
| MMAML+KML(ours) | – | 0.05 | 0.001 | 10 | 5 | 60000 |
| Multi-MAML | Grayscale | 0.4 | 0.001 | 10 | 1 | 60000 |
| | RGB | 0.01 | 0.001 | 4 | 5 | 60000 |
| MProtoNet | – | – | $0.001^\dagger$ | 10 | – | 30000 |
| MProtoNet+KML(ours) | – | – | $0.001^\dagger$ | 10 | – | 30000 |
| Multi-ProtoNet | Grayscale | – | $0.001^\dagger$ | 10 | – | 30000 |
| | RGB | – | $0.001^\dagger$ | 4 | – | 30000 |

# K    Information for checklist

**Limitations:** We have followed the procedures in [1] to construct multimodal datasets in our experiments for fair comparison with their work. Specifically, the following popular datasets have been used and we have also followed the combination procedures discussed in [1]: OMNIGLOT, MINI-IMAGENET, FC100, CUB, and AIRCRAFT. Potentially, additional datasets can be used in experiments to further demonstrate our ideas. Furthermore, our work has focused on few shot image classification. Our proposed ideas could be applicable to other few shot learning problems.

**Broader Impact:** Multimodal meta-learning is an extension of conventional few-shot meta-learning. Importantly, it mimics humans' ability to acquire a new skill via prior knowledge of a set of diverse skills. Research findings in this problem are very meaningful and important in machine learning. Furthermore, few shot classification studies the problem to classify samples from novel categories given only a few labeled data from each category. The setup is significantly different from other modern deep learning problems, but important for many domains where labeled data is difficult to obtain. For example, in clinical disease diagnosis, data needs to be labeled by medical experts and labeled data is expensive to obtain.

**Amount of compute:** All the results in this paper are produced by a machine with a single RTX 2080 Ti GPU. The amount of compute in this project is documented in Table 12. We follow submission guidelines to include the amount of compute for different experiments and CO2 emission.

Table 12: Amount of compute in this project. The GPU hours include computations for early explorations and experiments to produce the reported values. The carbon emission values are computed using `https://mlco2.github.io/`.

| Experiment | Hardware | GPU hours | Carbon emitted in kg |
|---|---|---|---|
| Multimodal Classification Results: Main paper Table 1 | RTX 2080 Ti | 480 | 51.84 |
| Unimodal Classification Results: Main paper Table 3 | RTX 2080 Ti | 36 | 3.89 |
| Visualization: Main paper Figure 1, Figure 2 and supplementary | RTX 2080 Ti | 5 | 0.54 |
| Hard Parameter Sharing: Supplementary Table 1 and Table 2 | RTX 2080 Ti | 90 | 9.72 |
| Verification of new-MMAML interpretation: Supplementary Table 3 | RTX 2080 Ti | 45 | 4.86 |
| Visualization: Supplementary t-SNE plot | RTX 2080 Ti | 10 | 1.08 |