# OpenReview forum: "Revisit Multimodal Meta-Learning through the Lens of Multi-Task Learning"
_NeurIPS.cc/2021/Conference — NeurIPS 2021 Poster_

### Official Review · Reviewer_uwYd · 2021-07-09

**Rating:** 6
**Confidence:** 4

**Summary:**

This paper draws a novel connection between multi-task learning (MTL) and few-shot classification as an instantiation of meta-learning. Plus, the paper suggests kernel modulation (KML), a drop-in replacement of FiLM, which can improve the performance of MMAML.

**Limitations And Societal Impact:**

I think the authors fairly well addressed the limitations and potential negative societal impact of their work.

**Main Review:**

This paper draws a novel connection between multi-task learning (MTL) and few-shot classification as an instantiation of meta-learning. Borrowing the term "multimodal" from their primary reference [^1], the authors treat modes as tasks. Using the concept of "transference" of MTL[^2], the authors also showed the usefulness of their proposed KML (Kernel Modulation) scheme, which increases the frequency of positive transfer.

Question 1. I see task embedding can improve the transference, as shown in figure 1. However, it is unclear that this change is originated from the use of KML, the other contribution point of the paper, or the use of task embedding. To verify that KML improves transference, I believe there should be a histogram for MprotoNet+FiLM for figure 1.

Comment 1. Although [^1] used the term "modality," I think the "domain" is the correct term for this paper's setting. Since FC100 and mini-ImageNet have natural image "modality" but have different input-label "domain."

KML is proposed to overcome the limitation of FiLM modulation, which MMAML[^1] used. KML directly alters the weight and bias of the convolution layer rather than alters its computed outputs. Since kernel weight requires more modulation parameters, the author used a 1-rank approximation (outer products). The authors report that this 1-rank approximation achieves higher meta-validation accuracy than the full-rank version in Figure 7 in the appendix.

Question 2. I am curious about the results from low-rank approximations other than 1-rank, for example, 2-rank or 3-rank.
Question 3. At figure 7, the authors stopped plotting after training iterations of 200K. Since the slope of full-rank MLP is higher than that of 1-rank, I guess full-rank meta-validation accuracy surpasses that of 1-rank after 200K. I believe a longer span of training iterations than 200K is needed to claim that 1-rank yields better results in terms of accuracy (line 210-211 in the appendix).

Comment 2. This KML scheme can be used as a drop-in replacement of FiLM. I believe a comparison with FiLM on the CLEVR dataset, which FiLM is originally evaluated on, can further validate the pros of KML.

Typos
- line 121, the subscript of theta
- line 138, met-learner
- line figure 1 caption, wrong capitalization "Proposed"

### Remarks
(Clarity) Overall, the paper is well written and easy to understand, though I want some parts to be fixed. (Typos)

[+] (Originality) Novel interpretation of multi-domain few-shot classification through the lens of transference.
[+] (Significance) Generalizable improvements on FiLM scheme, although it needs to be verified. (Comment 2)
[-] (Quality) I think the MTL analogy of the few-shot classification author made (transference) does not support the advantage of KML well. (Question 1)
[-] (Quality) The analysis of KML is insufficient. (Question 2 and 3)
[-] (Quality) The paper omits the RL experiment [^1] did. Since [^1] is its primary reference, the authors should conduct the RL experiment for the fairer comparison.

[^1] Vuorio, Risto, et al. "Multimodal Model-Agnostic Meta-Learning via Task-Aware Modulation." _ArXiv:1910.13616 [Cs, Stat]_, Oct. 2019. _arXiv.org_, [http://arxiv.org/abs/1910.13616](http://arxiv.org/abs/1910.13616).
[^2] Fifty, Christopher, et al. "Measuring and Harnessing Transference in Multi-Task Learning." _ArXiv:2010.15413 [Cs]_, Feb. 2021. _arXiv.org_, [http://arxiv.org/abs/2010.15413](http://arxiv.org/abs/2010.15413).

**Time Spent Reviewing:**

6 hours

---

> ### Author Response · Authors · 2021-08-10
> **[Responce for Reviewer uwYd] Part 2/2**
>
>
> > Question 3. At figure 7, the authors stopped plotting after training iterations of 200K. Since the slope of full-rank MLP is higher than that of 1-rank, I guess full-rank meta-validation accuracy surpasses that of 1-rank after 200K. I believe a longer span of training iterations than 200K is needed to claim that 1-rank yields better results in terms of accuracy (line 210-211 in the appendix).
>
> Thank you for your question. We really thank you for your time for going through the details in the supplementary as we believe a huge amount of our work is presented in the supplementary due to limitations in space. Actually, we have stopped the plotting because based on the numerical results during training, we saw no further improvement in full-rank MLP after 180K iterations. To make it clear we extended the span to 250K and updated the meta-validation figure. The expanded plot indicates that the mean meta-validation accuracy for both 200K and 250K epochs of single MLP is around 67\%, versus around 68\% in the proposed simplified structure.  Therefore, at 250K epochs, our proposed simplified structure still has better accuracy. The plot of a longer span will be included in the revised paper.
>
> To clarify this more, also as an example, the meta-test accuracy for the proposed simplified structure vs single MLP, in the case of 3Mode few-shot classification is shown below:
>
> | Method | 1-Shot | 2-Shot |
> | ------------- |:-------------:|:-------------:|
> | MLP      | 61.22$\pm$0.56\% | 69.38$\pm$0.48\% |
> | Proposed Simplified Structure      | **62.08$\pm$0.54\%** | **70.03$\pm$0.43\%** |
>
> We will add these details to the revised paper to clarify the better performance of the proposed simplified structure.
>
> $ $
>
> > Comment 2. This KML scheme can be used as a drop-in replacement of FiLM. I believe a comparison with FiLM on the CLEVR dataset, which FiLM is originally evaluated on, can further validate the pros of KML.
>
> Thank you for your suggestion. First, we remark that based on the proposed new interpretation of the FiLM scheme, our proposed KML can be seen as a generalization of FiLM. So we can expect that applying KML to the areas improved by FiLM may bring some further improvement.
>
> We also declare that the amount of improvement depends on the underlying structure of learning tasks. For example in the case of few-shot learning (especially multimodal distribution), since there could be a significant difference between different tasks (e.g., digit classification vs natural object classification), KML brings a large improvement over FiLM by letting the more powerful adaption of kernels for each few-shot task. Intuitively, this improvement may be less for the applications where more similar kernels are required for different tasks (e.g., visual reasoning on CLEVR dataset where there is a significantly lower variation on image statistics compared to our multimodal few-shot distribution, and the main difference originates from the question, and probably program signal).
>
> Based on your suggestion, we applied the KML to the CLEVR dataset by replacing KML with the FiLM in the [official code](https://github.com/ethanjperez/film) of the FiLM paper [1]. Following results are obtained:
>
> | Method       | Count | Exist |  Compare Numbers | Query Attribute | Compare Attribute| Average |
> | ------------- |:-------------:|:-------------:|:-------------:|:-------------:|:-------------:|:-------------:|
> | CNN+GRU+FiLM      | 94.3\% | 99.1\% | 96.8\% | 99.1\% | **99.1\%** | 97.7\% |
> | CNN+GRU+KML (ours)     | **96.1\%** | **99.5\%** | **97.1\%** | **99.3\%**| **99.1\%** | **98.2\%** |
>
>
> As the results show, KML on average improves the FiLM by 0.5% in the 5 question types. Considering the limited rebuttal time, we had not enough time to carefully tune the hyperparameters of KML. However, we are still able to obtain improvement using KML considering that FiLM has achieved very high accuracy already.
>
> $ $
>
> > Typos:
> line 121, the subscript of theta,
> line 138, met-learner,
> line figure 1 caption, wrong capitalization "Proposed"
>
>
> Thank you for your comments. We proofread the paper and fixed all Typos.
>
> $ $
>
> > **Remarks**:
> (Clarity) Overall, the paper is well written and easy to understand, though I want some parts to be fixed. (Typos)
>
> Thank you for your comment. We have proofread the paper to fix all Typos. As mentioned before, we will also add some comments to clarify probable ambiguities. Hope the applied changes address all your concerns.
>
> $ $
>
> > [+] (Originality) Novel interpretation of multi-domain few-shot classification through the lens of transference.
>
> We thank the Reviewer for the positive feedback.
>
> $ $
>
> >[+] (Significance) Generalizable improvements on the FiLM scheme, although it needs to be verified. (Comment 2)
>
> Thank you for your comment. The Reviewer’s positive feedback on our second contribution (KML) is very encouraging. We have provided the experimental results of KML on the CLEVR dataset and analysis of KML in our previous responses, and we will add these to the revised paper. Hope the provided details help to verify the generalizable improvement of the proposed KML on FiLM.
>
> $ $
>
> >[-] (Quality) I think the MTL analogy of the few-shot classification author made (transference) does not support the advantage of KML well. (Question 1)
>
> Thank you. As mentioned in previous responses, we added the histogram of MProtoNet to clarify the improvement of KML over FiLM in terms of knowledge transfer. Also, as mentioned before, since the transference metric is measuring the generalization performance, the results in Table1, Table 2, and Table 3 of the main paper indicate the better performance of KML in terms of knowledge transfer compared to FiLM in both cross-mode and within-mode scenarios. We will add these details to the revised paper. We hope the provided details clarify the advantage of KML and address your concerns.
>
> $ $
>
> > [-] (Quality) The analysis of KML is insufficient. (Question 2 and 3)
>
> Thank you for your comments. As mentioned before in our responses, we have added the detailed results of low-rank approximation based on your suggestion. Also, we have updated the meta-validation plot in supplementary together with adding the meta-test results to further justify the improvement brought by the proposed simplified structure. We have also provided some quick results of applying the KML to the CLEVR dataset (Visual Reasoning) based on your comment to validate the generalizability of this algorithm. We will add these details to the revised paper. We hope these details improve the analysis of the KML and address your concerns on this.
>
> $ $
>
> > [-] (Quality) The paper omits the RL experiment [^1] did. Since [^1] is its primary reference, the authors should conduct the RL experiment for the fairer comparison.
>
> Thanks for your comment. As mentioned in our supplementary, our work has focused on few-shot classification and we are able to perform more extensive classification experiments than [2] by considering more datasets and two popular types of meta-learners in our experiments, one of which (ProtoNet) is specifically designed for only few-shot classification. Note that [2] has focused on MAML only. We also remark that there are a lot of recent works in meta-learning literature that focus just on the few-shot classification due to its vital importance. Also, we remark that we have also made a considerable contribution to the analysis of knowledge transfer using transference.
>
> We believe our idea can be extended to RL (when using an optimization-based meta-learner like MAML). However, due to this short rebuttal period of one week, and also due to requirements of environmental setup for RL experiments, and several required modifications in the network structure, it is not feasible to provide the RL results at the moment. Our apology for that. We will work on RL for our final submission. Hope these explanations address your concerns.
>
>
> $ $
>
> We really thank Reviewer’s valuable time to review the submission and especially the Supplementary. We hope the rebuttal can help to increase the quality of our work. We sincerely hope that Reviewer can kindly re-consider our work (especially the quality of the analysis of the proposed KML and the connection to the proposed transference metric).
>
> $ $
>
> References:
>
> [1]    ​​Perez, Ethan, et al. "Film: Visual reasoning with a general conditioning layer." Proceedings of the AAAI Conference on Artificial Intelligence. Vol. 32. No. 1. 2018.
>
> [2]  Risto Vuorio, Shao-Hua Sun, Hexiang Hu, and Joseph J. Lim. Multimodal model-agnostic meta-learning via task-aware modulation. In Neural Information Processing Systems, 2019.
>
> [^1] used by Reviewer is the same as [2]

---

> ### Author Response · Authors · 2021-08-10
> **[Responce for Reviewer uwYd] Part 1/2**
>
> We thank the Reviewer for taking the time to review our paper and appreciate the valuable comments. The Reviewer’s opinion on Originality (our first contribution, transference analysis) and Significance (the performance of the proposed KML method) is valuable for us. Based on the comments from Reviewer, we will add the transference histogram for MProtoNet to show improvement of KML over FiLM considering the proposed transference metric. We will also provide more details on the generalization performance of the proposed simplified structure vs MLP to validate the improvement brought by our design. Finally, we will include the results of considering KML as a drop-in replacement of FiLM on the CLEVR dataset for visual reasoning. We hope these details and other responses address the concerns of the Reviewer.
>
> $ $
>
> > This paper draws a novel connection between multi-task learning (MTL) and few-shot classification as an instantiation of meta-learning. Borrowing the term "multimodal" from their primary reference [^1], the authors treat modes as tasks. Using the concept of "transference" of MTL[^2], the authors also showed the usefulness of their proposed KML (Kernel Modulation) scheme, which increases the frequency of positive transfer.
>
> Thank you for your comment. We appreciate your feedback on the novelty of our first contribution (adapting transference metric from MTL to episodic training of meta-learning) which tries to give a deeper understanding on the knowledge transfer between few-shot tasks from different modes (we have also tried to shed light on knowledge transfer in within mode scenario). Indeed the transference reveals the amount of negative transfer between few-shot tasks (specially cross-mode knowledge transfer), and this motivated us to propose KML to improve knowledge transfer. As you mentioned, the analysis results on both transference metric, and also the main meta-test classification results in the paper indicate the better performance of the KML compared to vanilla meta-learner and meta-learner+FiLM.
>
> $ $
>
> > Question 1. I see task embedding can improve the transference, as shown in figure 1. However, it is unclear that this change is originated from the use of KML, the other contribution point of the paper, or the use of task embedding. To verify that KML improves transference, I believe there should be a histogram for MprotoNet+FiLM for figure 1.
>
> Thank you for your comment. Apologies for not clarifying this. We will add the transference histogram of the MProtoNet (MProtoNet is exactly the same as our model, i.e. MProtoNet+KML, but with FiLM as a modulation scheme instead of our proposed KML) for the same source and target tasks to the revised paper to clarify this.
>
> Please note that in the transference histogram (we have produced the histogram already but cannot be included in OpenReview; we will add it to the revised paper), the following percentage of knowledge transfer are obtained:
>
> | Method       | Negative Transfer | Positive Transfer | Neutral |
> | ------------- |:-------------:|:-------------:|:-------------:|
> |ProtoNet  | 63% | 28% | 9% |
> |MProtoNet (FiLM), as suggested by Reviewer     | 64% | 26% | 10% |
> | MProtoNet+KML (ours)     | **55%** | **41%** | **4%** |
>
> The numbers show the improvement of the proposed KML over both ProtoNet and MProtoNet (FiLM). Also, please note that as the proposed transference is a metric of assessing generalization performance, the results in Table 1 for $2Mode^{\dagger}$ (miniImageNet and FC100) implicitly support the idea that improvement is brought by replacing FiLM with KML. Based on the results from Table 1 of the main paper, for the combination of mini-ImageNet and FC100, on average the performances of ProtoNet and MProtoNet is close together, and proposed MProtoNet+KML improves both of them. Better performance on KML for other mode combinations is also evident in Table 1 of the main paper.
>
> $ $
>
> > Comment 1. Although [^1] used the term "modality," I think the "domain" is the correct term for this paper's setting. Since FC100 and mini-ImageNet have natural image "modality" but have different input-label "domain."
>
> Thanks for considering this. We totally agree with you that the prevalent term used in the literature for referring to these multiple input-output combinations is the “multi-domain” term instead of “multi-modal”. However, please note that the idea behind using this term is the same as the MMAML paper which believes when using tasks from multiple domains, the task distribution $p(\mathcal{T})$ can be considered as a multimodal one. So, in order to make consistent with this work, we follow MMAML on using the “multimodal” term. In addition, in order to address your comment and prevent confusion, we will add the following explanation to the revised paper:
>
> “Note that multimodal meta-learning refers to multimodality occurs in task distribution $p(\mathcal{T})$ due to using tasks from multiple-domains, and should not be confused with multimodality in the data type (e.g., a combination of image, audio, and text).”
>
>
> $ $
>
> > KML is proposed to overcome the limitation of FiLM modulation, which MMAML[^1] used. KML directly alters the weight and bias of the convolution layer rather than alters its computed outputs. Since kernel weight requires more modulation parameters, the author used a 1-rank approximation (outer products). The authors report that this 1-rank approximation achieves higher meta-validation accuracy than the full-rank version in Figure 7 in the appendix.
>
> Thank you for your comment. Indeed since we need to generate a single parameter for each parameter within CNN, we have used this approximation, and surprisingly it worked better than full-rank in our system. To clarify, our intuition is that the main reasons for better performance of this simplified structure versus full-rank version are:
>
> 1. The main reason is reducing the risk of overfitting to meta-train classes using a simplified structure in which the number of parameters is 152 times less compared to the full-rank version (single MLP), see Supplementary for analysis of the number of parameters.
>
> 2. Another probable reason could be that since we are using an embedding vector of size 128 (like MMAML to have a fair comparison), the MLP incurs a more severe bottleneck compared to the proposed simplified structure.
>
>
> $ $
>
> > Question 2. I am curious about the results from low-rank approximations other than 1-rank, for example, 2-rank or 3-rank.
>
> Thank you for your interesting suggestion. We checked this idea by increasing the rank of the matrix with a simple idea. For producing the 2-rank approximation, we produce two different matrices: $M_{1}, and M_{2}$ using a similar method as (8) in our paper, and then add these two matrices to generate the final modulation matrix $M = M_{1} + M_{2}$. Please note that this time instead of 3 modules, we have 5 modules in our simplified structure. Two pairs of modules are used to generate the $M_{1}$ and $M_{2}$, and the fifth one is used to generate the bias term. We have checked these vectors to be independent. A similar procedure is used to design a 3-rank approximation of the MLP using three different pairs. Following results are obtained for 2Mode classification:
>
> | Setup       | MProtoNet | MProtoNet+KML(1-rank) | MProtoNet+KML(2-rank) | MProtoNet+KML(3-rank) |
> | ------------- |:-------------:|:-------------:|:-------------:|:-------------:|
> | 5-way, 1-shot | 70.60±0.56% | 73.69±0.52% | 72.12±0.54% | 72.06±0.52% |
> | 5-way, 5-shot | 75.72±0.47% | 79.82±0.40% | 78.94±0.43% | 78.70±0.46% |
>
> As results suggest, the 2-rank and 3-rank approximations still have better performance compared to the MProtoNet. However, the performance is degraded compared to the 1-rank approximation. The possible results could be overfitting of 2-rank and 3-rank versions due to more parameters.

---

> ### Comment · Reviewer_uwYd · 2021-08-23
> **Review update**
>
> I thank the authors for their detailed responses to my questions and comments.
>
> I'm raising my rating from 4 to 6 as my concerns are well addressed in the response, and the authors promised to perform RL experiments in their final version.

---

> > ### Author Response · Authors · 2021-08-30
> > **[Response for Reviewer uwYd]: RL Experiment results for the feedback “Review Update”**
> >
> > > I thank the authors for their detailed responses to my questions and comments.
> > I'm raising my rating from 4 to 6 as my concerns are well addressed in the response, and the authors promised to perform RL experiments in their final version.
> >
> > We really thank the Reviewer for the positive feedback. We have been working hard on the RL experiments and manage to finish the RL experiments. We are glad to report that **our proposed KML can achieve gain over [1] in all RL experiment setups**. We have followed strictly the RL experiment setup in [1]. We will include these RL experiment results in our final version. We will also release our code.
> >
> > More specifically, we have applied our KML algorithm on [the official code of MMAML](https://github.com/vuoristo/MMAML-rl) for RL experiments on three different environments used in [1]: Point Mass, Reacher, Ant. Similar to [1], for each environment, the goals are sampled from a multimodal goal distribution, with similar environment-specific parameters as [1]. To have a fair comparison, we have kept all other hyperparameters the same as the [1]. Following results are obtained for the mean and standard deviation of cumulative reward per episode for multimodal reinforcement learning problems with 2, 4 and 6 modes:
> >
> > - Results on **Point Mass 2D**
> >
> > | Method                       | 2 Modes              | 4 mdoes                | 6 Modes                |
> > | -------------                   |:-------------:            |:-------------:             |:-------------:             |
> > | MMAML                     | -136 $\pm$ 8        | -209 $\pm$ 32      | -169 $\pm$ 48       |
> > | MMAML+KML (ours) | **-121 $\pm$ 9**  | **-197 $\pm$ 30** | **-161 $\pm$ 41** |
> >
> > - Results on **Reacher**
> >
> > | Method                      | 2 Modes              | 4 mdoes                 | 6 Modes                |
> > | -------------                  |:-------------:            |:-------------:              |:-------------:             |
> > |MMAML                     | -10.0 $\pm$ 1.0   | -11.0 $\pm$ 0.8      | -10.9 $\pm$ 1.1     |
> > |MMAML+KML (ours) | **-9.6 $\pm$ 1.0**| **-10.6 $\pm$ 0.7**|**-10.6 $\pm$ 1.0**|
> >
> > - Results on **Ant**
> >
> > |Method                       | 2 Modes             | 4 Modes              |
> > | -------------                  |:-------------:           |:-------------:           |
> > |MMAML                     | -711 $\pm$ 25     | -904 $\pm$ 37    |
> > |MMAML+KML (ours) |**-689 $\pm$ 23**|**-891 $\pm$ 36**|
> >
> > $ $
> >
> > References:
> >
> > [1]  Vuorio, Risto, et al. "Multimodal Model-Agnostic Meta-Learning via Task-Aware Modulation." Advances in Neural Information Processing Systems 32 (2019): 1-12.

---

### Official Review · Reviewer_ScBy · 2021-07-10

**Rating:** 7
**Confidence:** 5

**Summary:**

- The paper analyses the important aspect of multimodal meta-learning in the context of few-shot classification. In this setting, the task distribution contains tasks sampled from multiple datasets with different input domains and labels,  introducing an extra layer of complexity.  It has been shown that sharing a single model allows for knowledge transfer across modes improving performance over unimodal training. However, it requires rethinking the training procedure since gradient updates from very different tasks could interfere with each other.
- In this regard, the authors propose a metric of "transference" to measure the positive or negative transfer across meta-learning tasks, based on a recently proposed approach for multi-task learning.
- They also propose a meta-learning model based on feature/kernel modulation.

**Limitations And Societal Impact:**

Limitations are expressed in the supplementary material. No negative societal impacts to declare in my opinion.

**Main Review:**

- One of the main issues that I found in this paper is that I'm not able to understand what is (are?) the real contribution(s). Most of the analysis is presented in the supplementary material and it's quite hard to follow the author's reasoning by reading just the main paper. This is something the authors should fix, but in my opinion, it would require substantial rewriting and restructuring the overall paper. My impression is that it lacks a proper structure that helps the reader understand the (huge) amount of work done by the authors. It might be the classic situation where a senior researcher with much more experience in writing could definitely improve the paper and make it ready for publication. Unfortunately, I don't think it is the case at the moment.
- To give an example, the transference measure and its analysis during training are nice, but the paper presents it as a means to develop a new algorithm for multimodal meta-learning. This is clearly not the case. In the next iteration, I would give more space to the analysis (supp B), while much less to the meta-learning model which is less important/marginal in my opinion. At the moment the two parts are simply stitched together, but there should be a more natural presentation.
- Another main point is that there's no really a comparison with other methods besides the MMAML paper (baseline). It's ok, but I think the authors may want to test their method with recent approaches. It would be nice to see results on the entire [meta-dataset](https://github.com/google-research/meta-dataset#1-triantafillou-et-al-2020) too. For other methods using conditional models, please see the [leaderboard](https://github.com/google-research/meta-dataset#leaderboard-in-progress). Besides the accuracy results, it would be nice to analyze the LR of other approaches. That would make the paper much more complete.

Other comments:
- **Lines 46-51.** I would rephrase: it should be clearer what is the main difference between Meta-Learning and MTL. The first optimizes the risk over future tasks sampled from a distribution of tasks. Note that this can be done because we are imposing a constraint on the types of tasks. Instead, MTL optimizes the average risk over a finite number of tasks.
- **Equations 4-5:** Following these equations I think the modulation is applied to the pre-activations. Is that right? In fact, 4-5 are true only if everything is linear. I was sure that modulations were generally applied after the nonlinearity, but checking [MMAML official code](https://github.com/shaohua0116/MMAML-Classification/blob/bdf1a93e798ab81619563038b95a3c5aa18717e0/maml/models/gated_conv_net.py#L16) it doesn't seem so. How do you use modulation in your model? Please clarify this point.
    - Anyway, I don't understand the point here. You are claiming that FILM (on the preactivation) is the same as scaling separately the i-th kernels (the convolutional filters) and biases. Again, no real surprise here if everything is linear. It shouldn't be verified empirically unless there is a nonlinearity somewhere and you want to show that the statement holds also in this case. What am I looking at here?
    - If I understood correctly your line of reasoning is that FILM modulation (scale and shift of the entire i-th feature, which corresponds to scale and shift of the i-th kernel) is not enough for multimodal meta-learning and you propose a method that scale and shift each element of the kernel (MKL), rather than using a single parameter (+ bias).
- Line 255: "Our experiments show that the proposed method decreases the number of parameters in g φ by a factor of 150..." wrt what? What are you comparing here? If you are comparing the naive way of producing MKL modulators with a single FC please be more precise. I don't see any real comparison here in the main paper. Is it all in supp D?. The authors should quantify the number of parameters of the models in the main paper or be more clear about it. Btw, the trick they propose to modulate each parameter/feature independently without exploding in the number of parameters is quite clever.
- Supp A (lines 35-37): **Hard parameter sharing**: I don't think this sentence is clear. What kind of parameters are you referring to? From my understanding, you consider hard sharing among the initialization of the weights that would be then adapted. This means that the initialization is not modulated before adaptation, hence they are shared. Is it correct? Please clarify because it might be confused with the ANIL approach of having shared parameters across tasks, namely, just there is not adaptation (no inner loop) and the parameters are just updated in the outer loop.
- Supp B (lines 96-98) **on generalization**: this is simply speculation unless you provide references/results supporting this.



————————————-

After the rebuttal and the discussion with authors and reviewers, I decided to increase my rate from 4 to 7.
The paper addresses an important open problem in meta-learning and provides a nice analysis from a multi-task perspective with empirical results. Hence, I recommend acceptance.
If accepted, the final version should include all the modifications agreed by the authors, and these should be verified by the reviewers and AC.

**Time Spent Reviewing:**

3.5h + 4h (after rebuttal)

---

> ### Comment · Reviewer_ScBy · 2021-08-10
> **Missing reference: Simon et al,  “On modulating the gradient  for meta learning”, ECCV 2020**
>
> Just an addendum to my review.
>
> Simon et al proposed to use the same (or similar) low-rank approximation (outer product) as KML to modulate gradients in gradient-based meta learners. Since your contribution is very similar to Simon et al, although applied to kernels rather than gradients, It think it would be appropriate to cite them and compare with their work.
>
> [Simon et al,  “On modulating the gradient  for meta learning”, ECCV 2020](https://www.ecva.net/papers/eccv_2020/papers_ECCV/papers/123530545.pdf)

---

> ### Author Response · Authors · 2021-08-10
> **[Response for Reviewer ScBy] Part 2/2**
>
>
> > Lines 46-51. I would rephrase: it should be clearer what is the main difference between Meta-Learning and MTL. The first optimizes the risk over future tasks sampled from a distribution of tasks. Note that this can be done because we are imposing a constraint on the types of tasks. Instead, MTL optimizes the average risk over a finite number of tasks.
>
> Thank you for your suggestion. We believe this is a well-articulated difference between meta-learning and MTL. We will rephrase this part on the revised paper, based on your suggestion:
> “... due to several fundamental differences between meta-learning and MTL (e.g, meta-learning optimizes the risk over the large number of future tasks sampled from an unknown distribution of tasks, while MTL optimizes the average risk over a finite number of known tasks; this will be further discussed).”
>
>
> $ $
>
> > Equations 4-5: Following these equations I think the modulation is applied to the pre-activations. Is that right? In fact, 4-5 are true only if everything is linear. I was sure that modulations were generally applied after the nonlinearity, but checking MMAML official code it doesn't seem so. How do you use modulation in your model? Please clarify this point.
>
> Thank you for your question. Please note that based on the official code of MMAML ([lines 97-36, and 145-159](https://github.com/shaohua0116/MMAML-Classification/blob/master/maml/models/gated_conv_net.py#:~:text=def%20conditional_layer(self,return%20x))), and also as explained in their supplementary, the modulation is applied after convolution operation and before non-linearity. This has also been clarified in section C (Figure 6) of our supplementary material.
>
> Answering your second question, as mentioned in (6) and (7), we directly modulate the model parameters including the weights (Convolution Kernels) and biases within each layer. We have also tried to clarify this difference in Figure 3 (parts (b) and (c)).
>
> $ $
>
> > Anyway, I don't understand the point here. You are claiming that FILM (on the preactivation) is the same as scaling separately the i-th kernels (the convolutional filters) and biases. Again, no real surprise here if everything is linear. It shouldn't be verified empirically unless there is a nonlinearity somewhere and you want to show that the statement holds also in this case. What am I looking at here?
>
> Thank you for your question. First, we show that FiLM is equivalent to scaling the whole elements of a kernel with a single number which limits the adaptation power of the modulated kernel to the assumed task. As explained in section C of supplementary, these are the same on pre-activation when the BN layer is bypassed. Actually, the empirical results are provided to also justify this during training when the Batch Normalization (BN) layer exists. Please note that in the official code of MMAML, the BN layer is applied before FiLM (Figure 6 in supplementary), and the affine transform of the BN layer is disabled, probably because a similar process is done by FiLM. Note that BN statistics in MMAML is computed based on the original feature maps (without modulation). However, in our interpretation, we apply BN on the feature maps which are generated by a modulated kernel. So, the statistics of the BN layer in our new interpretation are calculated based on the feature maps generated by modulated parameters. Due to this difference, we enable the affine transform of the BN layer in our new interpretation and compare the end-to-end training results to see whether this assumption holds when BN is used. Results show that from a training point of view, albeit differences in the BN layer, the original implementation (modulating feature maps) and the new interpretation (scaling kernels and biases as in equation (5) ) produce similar meta-test results.
>
> $ $
>
> > If I understood correctly your line of reasoning is that FILM modulation (scale and shift of the entire i-th feature, which corresponds to scale and shift of the i-th kernel) is not enough for multimodal meta-learning and you propose a method that scale and shift each element of the kernel (MKL), rather than using a single parameter (+ bias).
>
> Thank you for your comment. Yes, as discussed in the previous response, we believe that a major limitation of MMAML is that based on new interpretation the modulated kernel is produced by scaling the whole elements of the kernel by a single scalar number. To mitigate this issue, as shown in equations (6) and (7), our KML algorithm scales the whole elements of the kernel with a single number for each element (not shifting kernel elements) and adds a bias term to the bias vector of the whole layer. So, for each kernel, the convolution is applied using the modulated kernel and then the modulated bias term is added to produce the corresponding feature map.
>
> $ $
>
> > Line 255: "Our experiments show that the proposed method decreases the number of parameters in g φ by a factor of 150..." wrt what? What are you comparing here? If you are comparing the naive way of producing MKL modulators with a single FC please be more precise. I don't see any real comparison here in the main paper. Is it all in supp D?. The authors should quantify the number of parameters of the models in the main paper or be more clear about it. Btw, the trick they propose to modulate each parameter/feature independently without exploding in the number of parameters is quite clever.
>
> Apologies for the confusion. We mean comparing the simplified design for modulation parameters (figure 4) versus a single MLP. As you have mentioned, the detailed comparison is provided in the supplementary, and we just used this number as a pointer due to lack of space. We will modify this sentence in the revised paper to prevent further confusion:
> “Our experiments show that compared to a single MLP, the proposed simplified structure decreases the number of parameters in $g_{\phi}$ by a factor of 150 (Analysis of the number of parameters can be found in the Supplementary)”
>
> $ $
>
> > Supp A (lines 35-37): Hard parameter sharing: I don't think this sentence is clear. What kind of parameters are you referring to? From my understanding, you consider hard sharing among the initialization of the weights that would be then adapted. This means that the initialization is not modulated before adaptation, hence they are shared. Is it correct? Please clarify because it might be confused with the ANIL approach of having shared parameters across tasks, namely, just there is not adaptation (no inner loop) and the parameters are just updated in the outer loop.
>
> Thank you for raising this question. Actually we borrow the “hard parameter sharing” term from MTL literature.
>
> To make it more clear, for example, in the case of the first layer shared, we mean that we don’t apply the modulation for the parameters within this layer, meaning that parameters of the first layer are totally shared between all tasks and we do not generate the pseudo-task-specific parameters for this layer.
>
> Therefore, during meta-learning, the parameters of the first layer are not modulated (lines 5-7 of inner-loop in algorithm 2 will not be applied) and the initial parameters ($\theta$) are used for adaptation. Then in the outer loop, there would be only the update in line 10 (lines 11 and 12 will not be applied for the first layer).
>
> Additionally, since the remaining parameters (in the second, third and fourth layers) are not shared, we modulate the parameters of each of these layers using all steps in algorithm 2.
>
>
> $ $
>
> > Supp B (lines 96-98) on generalization: this is simply speculation unless you provide references/results supporting this.
>
> Thank you. Actually, this is just our intuition behind this behavior. We will change this part for clarification:
> “The probable reason could be that at the beginning ... ”
>
> $ $
>
> > **Just an addendum to my review.**
> Simon et al proposed to use the same (or similar) low-rank approximation (outer product) as KML to modulate gradients in gradient-based meta learners. Since your contribution is very similar to Simon et al, although applied to kernels rather than gradients, I think it would be appropriate to cite them and compare with their work.
> Simon et al, “On modulating the gradient for meta-learning”, ECCV 2020
>
> Thank you for pointing out this work, we will cite this work in the revised paper. We will also read the paper and get back.
>
> Our initial response: Our contribution in KML is that we develop a new interpretation of existing work MMAML, identify their limitation and propose a new modulation KML that can improve adaptability for multi-modal meta-learning. In order to avoid parameter explosion, we apply an outer product trick. In addition to the outer product trick, we also apply residual adjustment, see Eqn (6). We consider our insight into MMAML as our major effort in this part. We will read the paper suggested and will get back.
>
>
> $ $
>
> We really appreciate Reviewer’s valuable time to review the submission and especially the Supplementary. We hope the rebuttal can help clarify our work. We sincerely hope that Reviewer can kindly re-consider our work (especially the originality, technical quality and significance of our work).
>
> $ $
>
> References:
>
> [1]   Shengchao Liu, Yingyu Liang, and Anthony Gitter. Loss-balanced task weighting to reduce negative transfer in multi-task learning. In Proceedings of the AAAI Conference on Artificial Intelligence, volume 33,

---

> ### Author Response · Authors · 2021-08-10
> **[Response for Reviewer ScBy] Part 1/2**
>
> We sincerely appreciate Reviewer’s feedback. Reviewer’s description of our first contribution is very accurate: indeed we believe it is important to understand the training procedure in multimodal meta-learning and to analyze potential interference between tasks during gradient updates, therefore we propose a metric of “transference” inspired from multi-task learning.
>
> The analysis based on our proposed transference reveals that there are negative transfers among different tasks in the existing model. This finding motivates us to design a method Kernel ModuLation (KML, the second contribution mentioned by Reviewer) to reduce the negative transfer, using the idea of hard parameter sharing from MTL literature. To demonstrate the improvement in knowledge transfer using our proposed KML, we also use our proposed transference metric in addition to the conventional classification accuracy (e.g. Figure 1 and Supplementary, section B).
>
> We will make improvements to the revised paper to address each of the points you raised.
>
> $ $
>
> > One of the main issues that I found in this paper is that I'm not able to understand what is (are?) the real contribution(s).
>
> Thank you for your comment. Our apology for not making this clear enough. As mentioned in the introduction of the paper, we have two major contributions.
>
> First, in order to have a better understanding of knowledge transfer among different few-shot tasks, we have adapted the transference concept from the MTL and tailored it as a metric for analysis of the episodic training of the meta-learning.
>
> Second, to mitigate the negative transfer (as revealed by our transference analysis) and to help tasks learn from each other more (inspired by the hard parameter sharing from MTL, and also considering our new interpretation of modulation in MMAML) we propose the Kernel ModuLation (KML) scheme to decrease the negative transfer and improve generalization performance. Since KML requires generating a large number of parameters, we also propose a simple structure for parameter reduction on the generator.
>
> Furthermore, we apply the transference metric (our first contribution) to analyze our proposed KML (our second contribution), to demonstrate the improvement in knowledge transfer among tasks from different domains (e.g. Figure 1, and Supplementary).
>
> We will rephrase some sentences in the introduction to make our contributions more clear.
>
> $ $
>
> > Most of the analysis is presented in the supplementary material and it's quite hard to follow the author's reasoning by reading just the main paper. This is something the authors should fix, but in my opinion, it would require substantial rewriting and restructuring the overall paper. My impression is that it lacks a proper structure that helps the reader understand the (huge) amount of work done by the authors. It might be the classic situation where a senior researcher with much more experience in writing could definitely improve the paper and make it ready for publication. Unfortunately, I don't think that is the case at the moment.
>
> Thank you for your comment.  We really appreciate your time to go through the supplementary.
>
> As pointed out by the Reviewer, indeed there is “(huge) amount of work done” and it is not easy to present all of them in the main paper due to page limit, and we have to present a large amount of our analysis in the supplementary.
>
> Based on your suggestion, we added some more details about the major elements in the analysis part and tried to use more pointers to link the supplementary to the main paper. Furthermore, we plan to work on an extended version of this manuscript which we do not have page limit constraints. The following parts will be added to the analysis of transference in the main body of the revised paper:
>
> “Figure 2 indicates that in the cross mode knowledge transfer, the negative transference occurs at the beginning iterations, and increasingly more positive transference occurs as training proceeds. Similar behavior is observed in within-mode knowledge transfer (please see supplementary). Based on the experience from MTL literature, negative knowledge transfer occurs when different tasks fight for capacity [1]. In the next section, we will propose a new modulation scheme to reduce negative transfer and improve generalization (Figure 1). Additional transference analysis results on the reduction of negative transfer by the proposed method are deferred to the supplementary.”
>
> Please note that due to the amount of contribution in the KML part and also considerable interest in the KML part from other reviewers (e.g. Reviewer KUKC), we were not able to expand the analysis part that much in the main paper. Our apology for that. We sincerely hope that the added details could address your concern about reflecting the contribution in this part.
>
> Also, please note that one of the main takeaways from the analysis part is that when parameters are shared for all tasks, the negative transfer increases. Also, we have other important takeaways (mentioned in the main body of the paper) relating to knowledge transfer based on the similarity in the class or task hardness.
>
> $ $
>
> > To give an example, the transference measure and its analysis during training are nice, but the paper presents it as a means to develop a new algorithm for multimodal meta-learning. This is clearly not the case. In the next iteration, I would give more space to the analysis (supp B), while much less to the meta-learning model which is less important/marginal in my opinion. At the moment the two parts are simply stitched together, but there should be a more natural presentation.
>
> Thank you for your comment. As per our response to previous comments, the negative transference revealed by the proposed transference metric has motivated us to propose KML to mitigate this issue (KML is inspired by solutions in MTL literature and limitation of MMAML; this one is discussed in more detail in the first paragraph of section 5). Then we have applied the transference metric (our first contribution) to analyze our proposed KML (our second contribution), in particular, to understand the improved knowledge transfer due to KML at a micro-level via transference metric (e.g. Figure 1 and Supplementary). As per our previous response, we will add some details to the revised paper about this and hope the added part clarifies this.
>
> $ $
>
> > Another main point is that there's no really a comparison with other methods besides the MMAML paper (baseline). It's ok, but I think the authors may want to test their method with recent approaches.
>
> To the best of our knowledge, MMAML is the only available work for multimodal meta-learning. This is the reason that we have only compared our algorithm with MMAML and multiple designed baselines.
>
> $ $
>
> > It would be nice to see results on the entire meta-dataset too. For other methods using conditional models, please see the leaderboard. Besides the accuracy results, it would be nice to analyze the LR of other approaches. That would make the paper much more complete.
>
> Thank you for your suggestion. This is a really interesting suggestion. We have followed the experiment setup in the original MMAML paper and their datasets for direct comparison, but it would be useful to test on more datasets as we have also mentioned in “Limitations” (Supplementary Sec G).
>
> We are eager to try out the suggestion by the Reviewer. However, due to a very short period of rebuttal and the heavy requirements on data preparation for Meta-Dataset (210GB and one full day for pre-processing as mentioned in the official GitHub page of Meta-Dataset), the required memory and number of GPUs for handling this dataset (considering that we have used just one GPU), and several required modifications to adapt the task network to a varying number of classes, at the moment, we may not have enough time to obtain the results on the entire Meta-Dataset. But we will work towards that for final submission.
>
> $ $

---

> > ### Comment · Reviewer_ScBy · 2021-08-20
> > **Response to authors**
> >
> > Dear authors,
> > Thank you for your detailed responses.
> > I’ve accurately read another time your manuscript with you new insights and clarifications.
> >
> > As I’ve expressed to the other reviewers my main concern is that the contributions are not clearly stated and, in my opinion, the KML algorithm is a minor contribution. KML is mainly a (clever) trick to generate more modulation parameters without increasing model capacity, and it is similar to modgrad in terms of engineering, although the rationale is different. I encourage you to insert the main differences and similarities between KML and modgrad in the paper as listed in the response, giving less weight to its novelty.
> >
> > The new description of the contributions you provided in your response is what I expect to see in the final version. My suggestion, instead of claiming KML as a second contribution, is to restructure the second part of the paper and consider KML as a minor contribution by giving more importance to the analysis and comparison of FILM and KML from the the perspective of the transference metric you propose.
> >
> > In fact, you state that you **“apply the transference metric (our first contribution) to analyze our proposed KML (our second contribution), to demonstrate the improvement in knowledge transfer among tasks from different domains”**, but I don’t see any comparison to MMAML or FILM modulation in the paper nor in the supplementary in terms of transference metric.
> > You claim that KML has been developed to address FILM limitations in multimodal meta learning settings, so I think it is crucial to provide an analysis of the transference metric on protonet+FILM/MMAML and a comparison with your KML modulation variant in support of this claim (not just the final accuracy performance). In case I've missed this comparison, please point it to me and I will be more than happy to change my opinion.
> >
> > I believe that the paper would greatly benefit in terms of clarity and homogeneity by following my suggestions. I will continue to discuss with the other reviewers and AC, since this would require important changes in the paper structure.
> >
> > Minors misspell that I found in my recent reading:
> > - Figure 1: psitive -> positive
> > - line 138: met-learner -> meta-learner

---

> > > ### Author Response · Authors · 2021-08-20
> > > **[Response for Reviewer ScBy]: Regarding new feedback “Response to authors”**
> > >
> > > >Dear authors, Thank you for your detailed responses. I’ve accurately read another time your manuscript with your new insights and clarifications.
> > >
> > > We really thank the Reviewer’s time for re-reading the manuscript based on our responses.  And we really appreciate Reviewer’s suggestion.
> > >
> > > $ $
> > >
> > > > As I’ve expressed to the other reviewers, my main concern is that the contributions are not clearly stated and, in my opinion, the KML algorithm is a minor contribution. KML is mainly a (clever) trick to generate more modulation parameters without increasing model capacity, and it is similar to modgrad in terms of engineering, although the rationale is different.
> > >
> > > Thank you for your comment. We would like to emphasize that in the second part of our work we aim to find a solution to reduce negative transference in FiLM. We would like to respectfully mention that **our effort to uncover and pinpoint the weakness of FiLM via new interpretation (Line 184 - 198 in the main paper) is an important component of our contribution in this part.** Furthermore, we carefully consider the subtle effect of Batch Normalization to make sure that our new interpretation is accurate and our stated weakness of FiLM is correct (please kindly see our response to Reviewer’s previous question: “... It shouldn't be verified empirically unless there is a nonlinearity somewhere …”).
> > >
> > > Please note that **the weakness of FiLM as stated in our paper (Line 184 - 198) has not been discussed in other papers** to the best of our knowledge, and we think that **this weakness of FiLM uncovered in our paper is critical for multimodal few-shot learning when more adaptability is needed**. KML is proposed for more adaptability and the outer product version of it is a solution for preventing parameter explosion which interestingly performs better than a simple MLP solution. The gain obtained by addressing this weakness is rather substantial (e.g., as much as around **5%** for 5Mode few-shot classification). Furthermore, we can achieve gain by addressing this weakness of FiLM in another application, see our response to Reviewer uwYd. Hope this clarifies our effort in the second part of the paper.
> > >
> > > $ $
> > >
> > > > I encourage you to insert the main differences and similarities between KML and modgrad in the paper as listed in the response, giving less weight to its novelty.
> > >
> > > Thank you for your comment. To clarify the difference, we will add the difference between the ModGrad and the proposed KML (as listed in our previous response) into the revised paper.
> > >
> > > $ $
> > >
> > > > The new description of the contributions you provided in your response is what I expect to see in the final version. My suggestion, instead of claiming KML as a second contribution, is to restructure the second part of the paper and consider KML as a minor contribution by giving more importance to the analysis and comparison of FILM and KML from the perspective of the transference metric you propose.
> > >
> > > Thank you for the comment. As mentioned in our previous response to Reviewer, we will try to give more importance to the analysis and make more explicit links to the additional analysis in the Supplementary.
> > >
> > > $ $
> > >
> > > >In fact, you state that you “apply the transference metric (our first contribution) to analyze our proposed KML (our second contribution), to demonstrate the improvement in knowledge transfer among tasks from different domains”, but I don’t see any comparison to MMAML or FILM modulation in the paper nor in the supplementary in terms of transference metric. You claim that KML has been developed to address FILM limitations in multimodal meta-learning settings, so I think it is crucial to provide an analysis of the transference metric on protonet+FILM/MMAML and a comparison with your KML modulation variant in support of this claim (not just the final accuracy performance). In case I've missed this comparison, please point it out to me and I will be more than happy to change my opinion.
> > >
> > > Thank you for your comment. We totally agree with the Reviewer that this part is required to clarify more the performance of KML vs FiLM in terms of transference. Actually, in our initial response to Reviewer uwYd,  we have already generated the transference histogram to compare KML with FiLM, and we will add this part to our revised paper, as mentioned before:
> > >
> > > **Repeating our Response to Reviewer uwYd:** *“Apologies for not clarifying this. We will add the transference histogram of the MProtoNet (MProtoNet is exactly the same as our model, i.e. MProtoNet+KML, but with FiLM as a modulation scheme instead of our proposed KML) for the same source and target tasks to the revised paper to clarify this.*
> > >
> > > *Please note that in the transference histogram (**we have produced the histogram already but cannot be included in OpenReview; we will add it to the revised paper**), the following percentage of knowledge transfer are obtained:*
> > >
> > > | Method       | Negative Transfer | Positive Transfer | Neutral |
> > > | ------------- |:-------------:|:-------------:|:-------------:|
> > > |ProtoNet  | 63% | 28% | 9% |
> > > |MProtoNet (FiLM), as suggested by Reviewer     | 64% | 26% | 10% |
> > > | MProtoNet+KML (ours)     | **55%** | **41%** | **4%** |
> > >
> > >
> > > *The numbers show the improvement of the proposed KML over both ProtoNet and MProtoNet (FiLM). Also, please note that as the proposed transference is a metric of assessing generalization performance, the results in Table 1 for $2Mode^{\dagger}$ (miniImageNet and FC100) implicitly support the idea that improvement is brought by replacing FiLM with KML. Based on the results from Table 1 of the main paper, for the combination of mini-ImageNet and FC100, on average the performances of ProtoNet and MProtoNet are close together and proposed MProtoNet+KML improves both of them. Better performance on KML for other mode combinations is also evident in Table 1 of the main paper.”*
> > >
> > > We believe adding this comparison in terms of transference makes the advantage of the proposed KML more clear. Thanks for your suggestion.
> > >
> > > $ $
> > >
> > > > I believe that the paper would greatly benefit in terms of clarity and homogeneity by following my suggestions. I will continue to discuss with the other reviewers and AC, since this would require important changes in the paper structure.
> > >
> > > We really appreciate Reviewer’s valuable time to review and discuss our submission.
> > >
> > > $ $
> > >
> > > > Minors misspell that I found in my recent reading:
> > > Figure 1: psitive -> positive
> > > line 138: met-learner -> meta-learner
> > >
> > > Thank you for your comment. We have proofread the paper and fixed the Typos.

---

> > > > ### Comment · Reviewer_ScBy · 2021-08-21
> > > > **Response to authors 2**
> > > >
> > > > Thank you for your answers and for pointing to the discussion with rev uwYd.
> > > > The analysis with FILM is exactly what I was suggesting, sorry I’ve missed it.
> > > > I believe that it is quite an important point to be added to the narrative of the paper.
> > > >
> > > > I agree with most of comments from rev uwYd.
> > > > The paper is already packed of information so I don’t think that there is room for RL experiments in the current form, but I agree that it would have been a concrete set of experiments to show the benefits of KML and flaws of FILM.
> > > >
> > > > We will continue the discussion among reviewers and ACs.

---

> ### Author Response · Authors · 2021-08-12
> **[Response for Reviewer ScBy]: Regarding new feedback: "Missing reference: Simon et al, “On modulating the gradient for meta learning”, ECCV 2020".**
>
> > **Just an addendum to my review.**
> Simon et al proposed to use the same (or similar) low-rank approximation (outer product) as KML to modulate gradients in gradient-based meta learners. Since your contribution is very similar to Simon et al, although applied to kernels rather than gradients, I think it would be appropriate to cite them and compare with their work.
>
> **Repeating our initial response** (posted earlier): Our contribution in KML is that we develop a new interpretation of existing work MMAML, identify their limitations and propose a new modulation KML that can improve adaptability for multi-modal meta-learning. In order to avoid parameter explosion, we apply an outer product trick. In addition to the outer product trick, we also apply residual adjustment, see Eqn (6). We consider our insight into MMAML as our major effort in this part. We will read the paper suggested and will get back.
>
> **Adding to our initial response**: Please note that the only overlapping part of our work with ModGrad [1] is the outer product trick for parameter reduction in the generator. We will cite this paper in the revised version of our paper. Also, note that we have two main contributions in our paper: adapted transference analysis into episodic learning of meta-learning, and proposed KerneL Modulation (KML). In our second contribution, we expose the limitations of MMAML for multi-modal meta-learning and propose a more adaptable modulation (KML). Indeed, using a low-rank approximation is only a part of our second contribution.
>
> We compare our KML with ModGrad [1] for clarification. The followings are the **main differences** between the proposed KML and ModGrad :
>
> - While the proposed KML method modulates the kernels, ModGrad applies modulation on the gradients to surpass noisy gradients (the concept they claim arises due to limitation in data in few-shot scenarios).
>
> - From an embedding point of view, our KML maps extracted features from different tasks into different subspaces of the feature space. These spaces could be far from each other, enabling us to handle multimodal few-shot tasks generated from multiple different domains (digits, natural objects). However, ModGrad uses the same feature space for all tasks and modifies the loss space.
>
> - KML is a generalization of MMAML [3] (and specially FiLM [4]), while ModGrad can be considered as a task-aware version of Meta-SGD [5]. Please note that Meta-SGD extends MAML by learning Learning Rates (LRs)--a separate learning rate for each parameter. However, ModGrad can be considered as learning a generator to generate these LRs using a context vector (instead of learning these LRs directly).
>
> - Proposed KML modulates the parameters before the inner-loop update, so it generates a single modulation matrix for the whole adapting process in the inner-loop. However, ModGrad generates a different modulation matrix for the gradients in each step of the inner-loop update (Eqn (5) in [1]). For example, for optimization-based algorithms like MAML [2], and for 5 inner loop updates, we just generate a single modulation matrix to modulate the parameters before starting the inner-loop updates, while ModGrad generates five different modulation matrices, one for each step of the inner-loop update.
>
> - A substantial difference between the generator used in our KML and the one in ModGrad [1], is that like MMAML [3], we use embedding vectors (generated by task network) of each task for generating modulation parameters. However, ModGrad uses a zero-initialized seed as a context vector $\nu$ (input to generator) to generate an initial version of the modulation matrix. Then this context vector $\nu$ is updated using the loss from using the initial version of the modulation matrix. The updated context vector is then used to generate a more refined version of the modulation matrix. Note that this process is repeated for generating each modulation matrix in each iteration of the inner-loop. The parameters of the modulation matrix are updated in the outer-loop similar to KML.
>
> $ $
>
> References:
>
> [1]  Simon, Christian, et al. "On modulating the gradient for meta-learning." European Conference on Computer Vision. Springer, Cham, 2020.
>
> [2]   Finn, Chelsea, Pieter Abbeel, and Sergey Levine. "Model-agnostic meta-learning for fast adaptation of deep networks." International Conference on Machine Learning. PMLR, 2017.
>
> [3]   Vuorio, Risto, et al. "Multimodal Model-Agnostic Meta-Learning via Task-Aware Modulation." Advances in Neural Information Processing Systems 32 (2019): 1-12.
>
> [4] Perez, Ethan, et al. "Film: Visual reasoning with a general conditioning layer." Proceedings of the AAAI Conference on Artificial Intelligence. Vol. 32. No. 1. 2018.
>
> [5] Li, Zhenguo, et al. "Meta-sgd: Learning to learn quickly for few-shot learning." arXiv preprint arXiv:1707.09835 (2017).

---

### Official Review · Reviewer_KUKC · 2021-07-11

**Rating:** 8
**Confidence:** 4

**Summary:**

This work proposes a method to augment multi-modal meta learning by improving information transfer among tasks. In particular, a modulation matrix is generated for each layer and each task using the outer-product of two learned parameters. The meta learning algorithm then proceeds, but with the inner loop using the per-parameter modulation kernel to generate an output feature map. The proposed method improves positive information transfer among tasks as measured by transference and model performance when evaluated with two meta learning algorithms (MAML and ProtoNet) on several benchmark datasets.

**Limitations And Societal Impact:**

Societal impact is not mentioned, and limitations, notably increased latency/memory requirements by using -KML, should be analyzed to further strengthen the paper.

**Main Review:**

This paper is well-written and easy to follow. It builds on prior multi-modal meta learning work (MMAML). In particular, this work identifies a weakness in the design (that the modulated feature map effectively scales the original kernel by a scalar value), and suggests a per-parameter scaling kernel which appears to significantly improve empirical performance and information transfer among tasks.

The transference analysis lends explanation and intuition into why this method works. Moreover, adapting transference from the multitask domain to the meta learning domain is valuable and could be used in future algorithmic improvments to better understand the effect of proposed meta learning augmentations on positive/neutral/negative information transfer.

Given the robust meta learning transference analysis and strength of the empirical results, I believe this submission is technically sound. That said, I was unable to find any analysis related to the increased memory overhead or decrease in computational efficiency inherent in using this method. While this is perhaps a less important dimension of empirical analysis, this work would be strengthened if it was included.

Finally, multi-modal meta learning is an important domain with significant potential. Leveraging disparate modalities to improve model performance is/will be critical for many real world machine learning applications and research which develops this area is valuable.

For these reasons, I recommend this paper as an accept.

**Time Spent Reviewing:**

3.5

---

> ### Author Response · Authors · 2021-08-10
> **[Response for Reviewer KUKC]**
>
> We really thank the Reviewer for taking the time to review our paper. We appreciate Reviewer’s positive feedback on our work.
>
> In the revised manuscript, we will add discussion on analysis of memory overhead and computational efficiency to address your concern and strengthen our work.
>
> $ $
>
> > Given the robust meta-learning transference analysis and strength of the empirical results, I believe this submission is technically sound. That said, I was unable to find any analysis related to the increased memory overhead or decrease in computational efficiency inherent in using this method. While this is perhaps a less important dimension of empirical analysis, this work would be strengthened if it was included.
>
> Thank you for your comment. We will add the following details regarding the memory overhead and decreased computational efficiency of the proposed KML algorithm compared to the existing one (FiLM in MMAML [1]), and also a more general comparison of these in the context of multi-modal meta-learner:
>
> “In the main text, we have demonstrated that replacing FiLM with the proposed KML method significantly improves the accuracy of the meta-learner in both multimodal and conventional unimodal few-shot classification. KML achieves this substantial improvement by modulating the whole elements of the kernel instead of applying the affine transform on the feature maps (FiLM in [1]). This is done by generating a larger number of parameters compared to FiLM. Here we analyze the overhead introduced by replacing FiLM (existing method in [1]) with KML (our proposed method).
>
> First, we discuss the number of additional parameters introduced by KML. Since the only difference between the two methods is on the generator, we consider this module for comparison. Recalling from section D of the supplementary, the number of parameters required in proposed simplified structure in KML (for a layer) is $N_{\upsilon} \times (N_{k} \times N_{k} \times N_{i} + 2N_{o})$. Since in FiLM, only two parameters are generated for each channel of the convolutional layer, the number of parameters in the generator is $N_{\upsilon} \times (2N_{o})$. Therefore, the additional overhead of KML for the generator becomes $N_{\upsilon} \times N_{k} \times N_{k} \times N_{i}$ for each layer. Considering we have four convolutional layers in our structure and for each layer, a separate generator is used, in total, KML adds around 261.5 K parameters to the ones in FiLM. Considering this number, for example, the total number of parameters in MProtoNet+KML (our method) increases by 22.9\% compared to MProtoNet (existing method in [1]).
>
> Second, in terms of computational overhead, the following table shows the total training time for MProtoNet (existing method in [1]) and MProtoNet+KML (ours) for 2Mode (combination of Omniglot and miniImageNet), 5-way 5-shot scenario:
>
> | Method        | Training Time    | Accuracy  |
> | ------------- |:-------------:|-------------:|
> | MProtoNet      | 173 minutes | 56.03$\pm$0.64\% |
> | MProtoNet+KML      | 182 minutes     |   59.31$\pm$0.62\% |
>
> As the results show, the computational overhead of the proposed method in training time is around 5.2\%. Similar training results are obtained for the other setups (3 Mode, 5Mode). Also please note that the inference time of our method and existing method [1] are almost the same (on average 0.087 seconds for each mini-batch of few-shot tasks).”
>
> $ $
>
> > Societal impact is not mentioned, and limitations, notably increased latency/memory requirements by using -KML, should be analyzed to further strengthen the paper.
>
> Thank you for your comment. We hope our previous response will address your concern about the memory requirements and increased latency using our proposed KML, and we will include that in the revised manuscript.
>
> $ $
> $ $
>
> References:
>
> [1]  Risto Vuorio, Shao-Hua Sun, Hexiang Hu, and Joseph J. Lim. Multimodal model-agnostic meta-learning via task-aware modulation. In Neural Information Processing Systems, 2019.

---

> > ### Comment · Reviewer_KUKC · 2021-08-10
> > **Updated Review**
> >
> > The author's response has addressed my questions/comments, and after rereading the paper in the context of this response as well as the public discussion among other reviews and the authors, I've increased my rating of this work. My reasoning is below:
> >
> > 1. **Contribution:** First, this work analyzes transfer dynamics in meta-learning, with a focus on transfer across different modalities by examining optimization dynamics. This could be a paper in itself, and I have not seen past work analyze this dimension of meta learning to this extent. Second, they propose a novel kernel modulation method which substantively improves upon prior work.
> >
> > 2. **Relevance:** Multi-modal learning as well as contrastive learning are two domains that have seen tremendous progress in 2021 to manifest astounding results. This work takes a step in multi-modal learning to improve upon existing meta learning methods. In short, I believe the transfer learning dynamics analysis among modalities as well as the proposed KML method itself are highly relevant and will be used in both industry and academia.
> >
> > 3. **Analysis:** The analysis is well written, motivated, and thorough. I believe the comparison systems are appropriate as well as the benchmark datasets to demonstrate the efficacy of KML. Moreover, the analysis into positive/negative transfer adds a dimension of explainability to their method, and the generator network does not add prohibitive (or even substantive) overhead to their method.
> >
> > In summary, I think this is a superior paper that makes a meaningful contribution to the field of meta learning.

---

### Official Review · Reviewer_79Fb · 2021-07-11

**Rating:** 7
**Confidence:** 3

**Summary:**

This paper introduces a novel few-shot multitask learning approach that uses an additional kernel modulation module that computes the residual adjustment matrix for a given pair of tasks. Authors propose an approach for analyzing the knowledge transfer of particular tasks on the accuracy of the other tasks based on transference.  Authors demonstrate efficiency of the proposed method in 5-shot and 1-shot setups on Omniglot, mini-ImageNet, FC100 and other datasets and compare their result with the MAML and ProtoNet baselines.


**Limitations And Societal Impact:**

The limitations are not properly discussed (see the last paragraph of the main review).

**Main Review:**

The paper is overall well-written and easy to understand, except the introduction, which needs to be more to the point.
Authors introduced a new way of analyzing how training the model with specific subtasks (called episodes in the paper) affects its performance on other subtasks at any stage of training based on the losses before and after training, which is a reasonable idea that can be useful to the multitask learning community. My only concern is that in Figure 2, it is shown that the amplitude of transference decreases with the number of epochs, which might be caused by the learning rate decay. It is overall not surprising that the variance in predictions - and subsequently the amplitude of transference - decreases with time. More experiments are needed to exclude all these factors from the analysis.


The overall idea of inferring the task-specific model weights is not new [a. “TRACE NORM REGULARIZED DEEP MULTI-TASK LEARNING”, Yang et al., ICLR 2017; b. “DEEP MULTI-TASK REPRESENTATION LEARNING: A TENSOR FACTORISATION APPROACH”m Yang et al.,  ICLR 2017; c. “Attentive Single-Tasking of Multiple Tasks”, Maninis et al., CVPR 2019, and others], and these should be added to the related work section. However, authors are the first to efficiently apply this idea to few-shot multitask learning without a significant overhead in the number of parameters, which is a valuable contribution to the community.


The experimental evaluation as well as the transference results clearly indicate that incorporation of the proposed kernel modulation increases the overall accuracy on the new tasks in all setups, both for the MAML and ProtoNet baselines.


The tradeoffs and limitations of the proposed approach are not properly discussed. Is it possible that the additional modules g_\phi result in overfitting to the existing tasks? What is the trade-off in the number of parameters and training time and the accuracy of the proposed method? These should be analyzed as well.


Minor:
a) Is there a practical reason for using separate embedding modules g_\phi_1 and g_\phi_2 instead of a single module?
b) L255-257: Probably the authors meant “replacing MAML with ProtoNet”, not the other way around.


**Time Spent Reviewing:**

2

---

> ### Author Response · Authors · 2021-08-10
> **[Response for Reviewer 79Fb]**
>
> We sincerely appreciate the Reviewer’s positive feedback on the novelty and significance of our work. We will add some details to include the memory and computational overhead of the proposed method. We will also discuss your comment regarding the dependency of the proposed transference metric on the learning rate. We hope the provided responses address your concerns.
>
> $ $
>
> > The paper is overall well-written and easy to understand, except the introduction, which needs to be more to the point.
>
> Thank you, we will rephrase some sentences in the introduction to make them more to the point.
>
> $ $
>
> > My only concern is that in Figure 2, it is shown that the amplitude of transference decreases with the number of epochs, which might be caused by the learning rate decay. It is overall not surprising that the variance in predictions - and subsequently the amplitude of transference - decreases with time. More experiments are needed to exclude all these factors from the analysis.
>
> Thank you for this very interesting comment.
>
> First, we remark that we have followed the standard training procedures to reflect the exact training dynamics for these models, and that is why learning rate decay is applied.
>
> Indeed Reviewer is correct that the learning rate decay to some extent may cause the decrease in amplitude of transference as we observe in Figure 2.
>
> On the other hand, the main findings in our paper are based on **the number/percentage of positive/negative transfers**, and this should be less dependent on the learning rate.
>
> More specifically, we use equation (2) loss ratio (LR) to measure transference. Based on that, a **positive** transference occurs if the loss of a task $j$ **decreases** after applying a gradient update on the parameter (w.r.t. to a task $i$), and equivalently $LR_{i \rightarrow j} < 1$. This would depend mostly on the direction of the parameter update as long as the step size is small enough (when a suitable learning rate is used). Therefore, the polarity (positive/negative) of the transference would be less affected by the learning rate decay.
>
> In this work, we show that our proposed MProtoNet+KML has led to a higher percentage of positive transference compared to the existing method ProtoNet. In Figure 2, we can observe from the histogram that there is a higher percentage of positive transference as the training proceeds. We make these remarks based on the polarity of the transference.
>
> But Reviewer has made an excellent point. In the revised paper, we will make this remark so that the transference will be properly used by others. In particular, if the amplitude of transference is used in other analyses, one would need to pay attention to the learning rate decay.
>
> $ $
>
> > The overall idea of inferring the task-specific model weights is not new [a. “TRACE NORM REGULARIZED DEEP MULTI-TASK LEARNING”, Yang et al., ICLR 2017; b. “DEEP MULTI-TASK REPRESENTATION LEARNING: A TENSOR FACTORISATION APPROACH”m Yang et al., ICLR 2017; c. “Attentive Single-Tasking of Multiple Tasks”, Maninis et al., CVPR 2019, and others], and these should be added to the related work section.
>
> Thank you, we will add these papers to the related works.
>
> $ $
>
> > However, authors are the first to efficiently apply this idea to few-shot multitask learning without a significant overhead in the number of parameters, which is a valuable contribution to the community.
>
> We really thank the Reviewer for the positive feedback.
>
> $ $
>
> > The tradeoffs and limitations of the proposed approach are not properly discussed. Is it possible that the additional modules $g_\phi$ result in overfitting to the existing tasks?
>
> Thank you for raising this question. Please note that by applying our simplified structure (Figure 4), the number of required parameters in this module $g_\phi$ is significantly reduced (152x compared to single MLP in the original design) which in turn reduces the risk of overfitting. Also, since the generator module is shared among few-shot tasks from multiple different modes, we believe this lowers the risk of overfitting, since it uses multiple tasks from different domains, and based on multi-task learning literature this lowers the risk of overfitting [1, 2].
>
> $ $
>
> > What is the trade-off in the number of parameters and training time and the accuracy of the proposed method? These should be analyzed as well.
>
> Thank you for your comments. Apologies that we have not made this one clear. There are some details on comparing the proposed simplified structure with a single MLP in the supplementary. We will also add the following details in the revised paper, to address the parameter and computational overhead of KML over previous modulation (FiLM) used in the MMAML [3]:
>
> “In the main text, we have demonstrated that replacing FiLM with the proposed KML method significantly improves the accuracy of the meta-learner in both multimodal and conventional unimodal few-shot classification. KML achieves this substantial improvement by modulating the whole elements of the kernel instead of applying the affine transform on the feature maps (FiLM in [3]). This is done by generating a larger number of parameters compared to FiLM. Here we analyze the overhead introduced by replacing FiLM (existing method in [3]) with KML (our proposed method).
>
> First, we discuss the number of additional parameters introduced by KML. Since the only difference between the two methods is on the generator, we consider this module for comparison. Recalling from section D of the supplementary, the number of parameters required in proposed simplified structure in KML (for a layer) is $N_{\upsilon} \times (N_{k} \times N_{k} \times N_{i} + 2N_{o})$. Since in FiLM, only two parameters are generated for each channel of the convolutional layer, the number of parameters in the generator is $N_{\upsilon} \times (2N_{o})$. Therefore, the additional overhead of KML for the generator becomes $N_{\upsilon} \times N_{k} \times N_{k} \times N_{i}$ for each layer. Considering we have four convolutional layers in our structure and for each layer, a separate generator is used, in total, KML adds around 261.5 K parameters to the ones in FiLM. Considering this number, for example, the total number of parameters in MProtoNet+KML (our method) increases by 22.9\% compared to MProtoNet (existing method in [3]).
>
> Second, in terms of computational overhead, the following table shows the total training time for MProtoNet (existing method in [3]) and MProtoNet+KML (ours) for 2Mode (combination of Omniglot and miniImageNet), 5-way 5-shot scenario:
>
> | Method        | Training Time    | Accuracy  |
> | ------------- |:-------------:|-------------:|
> | MProtoNet      | 173 minutes | 56.03$\pm$0.64\% |
> | MProtoNet+KML      | 182 minutes     |   59.31$\pm$0.62\% |
>
> As the results show, the computational overhead of the proposed method in training time is around 5.2\%. Similar training results are obtained for the other setups (3 Mode, 5Mode). Also please note that the inference time of our method and existing method [3] are almost the same (on average 0.087 seconds for each mini-batch of few-shot tasks).”
>
> $ $
>
> > Minor: a) Is there a practical reason for using separate embedding modules g_\phi_1 and g_\phi_2 instead of a single module?
>
> Our Response: Thank you for your question. Actually as mentioned in the main paper, the reason is to reduce parameters in the generator. We have shown this structure reduces the number of required parameters in the generator by a factor of 152x compared to the original design (single MLP; please see Supplementary). This parameter reduction allows us to prevent the parameter explosion in the model, train the model faster, and also prevent the overfitting of the generator. So we can have better generalization on the meta-test classes. Our meta-test results (will be added to the supplementary) on 3Mode classification shows that this parameter reduction surprisingly improves the meta-test accuracy by an average of 0.7\%:
>
> | Method        | 1-Shot   | 5-Shot |
> | ------------- |:-------------:|-------------:|
> | MLP     | 61.22$\pm$0.56\% | 69.38$\pm$0.48\% |
> |Proposed Structure    | **62.08$\pm$0.54\%**   |   **70.03$\pm$0.43\%**|
>
> Similar improvements are observed for other setups (2Mode, 5Mode).
>
> $ $
>
> > b) L255-257: Probably the authors meant “replacing MAML with ProtoNet”, not the other way around.
>
> Thank you. We will change the order of MMAML and the MProtoNet phrases in the text to prevent confusion:
>
> “Comparing MProtoNet with MMAML, we can see that replacing ProtoNet with MAML considerably improves the classification accuracy.”
>
> $ $
>
> >The limitations are not properly discussed (see the last paragraph of the main review).
>
> Thank you. We will add the details of parameters and computational overhead to the revised manuscript to address your concern about this.
>
> $ $
>
> References:
>
> [1]    Zhang, Yu, and Qiang Yang. "A survey on multi-task learning." IEEE Transactions on Knowledge and Data Engineering (2021).
>
> [2]  Ruder, Sebastian. "An overview of multi-task learning in deep neural networks." arXiv preprint arXiv:1706.05098 (2017).
>
> [3]  Risto Vuorio, Shao-Hua Sun, Hexiang Hu, and Joseph J. Lim. Multimodal model-agnostic meta-learning via task-aware modulation. In Neural Information Processing Systems, 2019.

---

> > ### Comment · Reviewer_79Fb · 2021-08-18
> > **Follow-up comment**
> >
> > After reading the responses provided by the authors, as well as the reviews, I am thoroughly convinced that this is a good paper that deserves to be accepted to the venue.
> >
> > Almost all the questions and concerns raised in my review were properly addressed in the rebuttal. Thank you for clarifying my misunderstanding regarding the transference metric. The discussion of the parameter efficiency/accuracy/training time in the response above is sufficient and should be added to the main paper. I am still not convinced about separate embedding modules g_\phi_1 and g_\phi_2, why would two embedding modules be more parameter-efficient than one?
> >
> > I do not agree with the feedback from Reviewer ScBy. In my opinion, the paper is very well-written and clarity is not an issue. The fact that the proposed transference analysis approach is not directly related to the proposed method is also not a sufficient reason to reject the paper. I also find MAML an appropriate choice for a baseline in this case.

---

> > > ### Author Response · Authors · 2021-08-19
> > > **[Response for Reviewer 79Fb]: Regarding new feedback “Follow-up Comment”**
> > >
> > > > Almost all the questions and concerns raised in my review were properly addressed in the rebuttal. Thank you for clarifying my misunderstanding regarding the transference metric. The discussion of the parameter efficiency/accuracy/training time in the response above is sufficient and should be added to the main paper.
> > >
> > > We appreciate the Reviewer’s positive feedback. We will add the discussion of the parameter efficiency/accuracy/training time into the revised paper.
> > >
> > > $ $
> > >
> > > > I am still not convinced about separate embedding modules g_\phi_1 and g_\phi_2, why would two embedding modules be more parameter-efficient than one?
> > >
> > > Thank you for your comment. Sorry for not clarifying this properly. Actually, as mentioned in the general framework, $g_\phi$ is the parameter generator. As we need to modulate each element in the kernel, we need to generate a very high number of parameters. And generating this number of parameters using a single MLP requires an MLP with a very large number of parameters (**49.7 Million parameters** in total as mentioned in Section D of supplementary). This increases the network size, reduces the training speed, and increases the chance of overfitting to meta-train classes. Therefore we use the outer product trick (Figure 4) to reduce the number of parameters in the generator. Now, instead of **a large MLP**, we have **three smaller MLPs** as generators (**327K parameters** in total). The point is that using smaller MLPs $g_{\phi_1}$ and $g_{\phi_2}$, we generate a smaller number of parameters (compared to one large MLP). But, then we use the **outer product** operation (Eqn (8)) to expand these parameters and provide the network with the required number of parameters.
> > >
> > > As mentioned before, this structure reduces the number of parameters in the generator with a factor of 152, increases the training speed, and reduces the chance of overfitting (which results in better generalization as mentioned in the previous response). We hope this clarifies the idea behind using the proposed simplified structure (with smaller MLPs) instead of a single large MLP.

---

### Comment · Reviewer_ScBy · 2021-08-29
**Request to the authors: summary of the updates to appear in the final paper**

Dear authors,

Thank you for all your answers and your active participation in the discussion.

In order to help the reviewers and the AC to take the final decision, I suggest you summarize in a single comment a detailed list of all changes and clarifications that you intend to incorporate into the paper.
This will help us track the differences between the initial and final versions of the paper and verify that these changes are actually happening in case of acceptance. Also, I think this would be particularly helpful for the AC and a good way to evaluate the impact of the review process for this specific paper.

Thanks a lot!

---

> ### Author Response · Authors · 2021-08-31
> **[Response for Reviewer ScBy]: Regarding new feedback “Request to the authors: summary of the updates to appear in the final paper”**
>
> > Dear authors,
> >Thank you for all your answers and your active participation in the discussion.
> >In order to help the reviewers and the AC to take the final decision, I suggest you summarize in a single comment a detailed list of all changes and clarifications that you intend to incorporate into the paper. This will help us track the differences between the initial and final versions of the paper and verify that these changes are actually happening in case of acceptance. Also, I think this would be particularly helpful for the AC and a good way to evaluate the impact of the review process for this specific paper.
> >Thanks a lot!
>
> We really appreciate Reviewer’s valuable time to review and discuss our submission. Here we provide a summary of all changes that we incorporate into the paper. For each change, we have also mentioned the Reviewer(s) addressed it.  The following changes  will be applied to the final version:
>
> 1. We will update the description of our contributions as we responded to Reviewer ScBy, i.e. we will give more importance to the discussion of transference analysis, will highlight the use of transference metric to analyze and compare FiLM and KML, and will include pointers to explicitly link to the discussion in the Supplementary.  In order to include more discussion of transference analysis, some experiments of KML will be moved to Supplementary (e.g. unimodal few-shot classification experiments). [Reviewer ScBy]
>
> 2. We will clarify the followings by adding more details: a) “hard-parameter sharing” term used in Supp A, b) the difference between MTL and meta-learning, c) usage of “multimodal term” instead of “multi-domain”, d) the differences between proposed KML and ModGrad [1] algorithm, and e) adding more works from MTL literature to related work to cover it better.  [Reviewers ScBy, uwYd, and 79Fb]
>
> 3. We will include a detailed discussion of the increased memory overhead and computation complexity using the proposed KML versus FiLM to clarify the better performance of the proposed simplified structure. [Reviewers KUKC and 79Fb]
>
> 4. We will modify the meta-validation figure (in Supp), and include the meta-test results of the proposed simplified structure vs a single MLP. This will clarify more on the better performance of the proposed structure vs a single large MLP. [Reviewer uwYd]
>
> 5. In order to verify the generalizability of KML, we will include the experimental results of applying the KML on the original FiLM paper [2] (visual reasoning), and also the reinforcement learning results. We will also include the results of rank experiments. Note that we have completed these experiments and have posted the results already. [Reviewer uwYd]

---

### Decision · Program_Chairs · 2021-09-27

**Decision:**

Accept (Poster)

**Comment:**

The submission tackles the problem of meta-learning on a multimodal task distribution. It introduces an analytical methodology inspired from recent work on transference in multi-task learning and proposes a new multimodal meta-learning approach called Kernel Modulation (KML) which is claimed to outperform competing approaches.

Reviewers found the problem tackled to be important to the research community and the transference analysis to be an interesting and valuable contribution borrowed from the multi-task learning literature. They expressed reservations about the way in which the contributions were framed (in that the transference analysis feels disconnected from the KML contribution) and how many important details are relegated to the supplementary material. However, given that the authors were very responsive and open to incorporating their feedback, the reviewers and I feel positive about acceptance, if the changes promised are incorporated in the final version (see summary here: https://openreview.net/forum?id=V5prUHOrOP4&noteId=l5jMUHLSraH).